# Stochastic parametric skeletal dosimetry model for humans: Pediatric and adult computational skeleton phantoms for internal bone marrow dosimetry

Pavel A. Sharagin[1], Elena A. Shishkina[1,2], Evgenia I. Tolstykh[1], Michael A. Smith[3]*, Bruce A. Napier[3]

1 Urals Research Center for Radiation Medicine, Chelyabinsk, Russia, 2 Chelyabinsk State University, Chelyabinsk, Russia, 3 Pacific Northwest National Laboratory, Richland, Washington, United States of America

* Michael.Smith@pnnl.gov

## Abstract

Currently, computational phantoms that simulate skeletal tissues are used in active red bone marrow (AM) internal dosimetry. Up-to-date reference computational phantoms recommended by the ICRP are based on the analysis of CT-images of cadavers. Such phantoms have significant disadvantages. One disadvantage is that the assessment of uncertainty due to the population variability of skeleton dimensions and microstructure results from the limited availability of autopsy material. Another disadvantage is the simplified modelling of cortical layer and bone microarchitecture. A method of stochastic parametric skeletal dosimetry modelling of the bone structures – SPSD modelling – has been developed as an alternative to the ICRP reference phantoms. In the framework of this approach, skeletal phantom parameters are evaluated based on extensively reviewed results of published measurements of real bones. The SPSD approach allows for the assessment of both population-average values and their variability. SPSD-phantoms of the skeleton are modelled in voxel representation. They consist of smaller phantoms of the bone sites – segments – described by simple geometric shapes with uniform microarchitecture parameters. Such segmentation makes it possible to account for non-homogeneous skeletal microarchitecture and to model the bone structure with the required voxel resolution to elaborate suitable skeletal phantoms. The current study presents the parameters of the SPSD skeletal phantoms for the following age-groups: newborn, 1-year-old, 5-year-old, 10-year-old, 15-year-old (male and female), and adult (male and female). This skeletal phantom can be used for dosimetry as an alternative to available reference phantoms for bone-seeking radionuclides. The above-mentioned age- and sex-specific skeletal phantoms are comprised of 289 unique segments. The characteristics of the SPSD phantoms do not contradict published data and are in good agreement with the measurement results of real bones.

**Data availability statement:** All relevant data are within the manuscript and its Supporting information files.

**Funding:** This work was funded by Federal Medical-Biological Agency of Russia (PAS, EAS, EIT) and the U.S. Department of Energy's Office of International Health Programs (MAS, BAN) in the framework of joint US-Russia JCCRER Project 1.1 (https://www.energy.gov/ehss/russian-health-studies-program).

**Competing interests:** The authors have declared that no competing interests exist.

## Introduction

Measuring the distribution of radiation doses inside a living organism using physical methods is often impossible for several reasons. First, the distribution of doses depends on the relative position of the radiation source and the target organ; then, each type of radiation has its own characteristic principles of interaction with media and penetrating power. But more importantly, the human body consists of many complex anatomical structures, heterogeneous in density and composition [1].

Computational human phantoms are computer models used to obtain dose distributions within a human body exposed to internal or external radiation sources. They are also used to model the detector response of in vivo whole-body counters (WBCs) [2]. Over the past two decades, there has been a significant increase in activity in the field of modeling of a "digital human" to create dosimetric computational phantoms [3]. Phantoms have become one of the necessary tools of computational dosimetry today. There are 2 directions of technology development for constructing computational human phantoms: (1) modeling of a reference individual [4,5] and (2) creation of personalized phantoms [3]. The first direction is highly relevant in radiation protection, dose reconstruction and dose prediction for workers and members of the public and personnel, as well as for uniformity in the numerical calibration of WBCs. The second is required in diagnostic radiology and nuclear medicine for individualized treatment and evaluation. In this paper, we focus on the 1st direction.

ICRP defined reference anatomical descriptions for male and female subjects of six different ages: newborn, 1 year, 5 years, 10 years, 15 years, and adult [4,6,7]. In selecting reference ages, ICRP used data on Western Europeans and North Americans because these populations have been well studied with respect to anatomy, body composition, and physiology [8]. Descriptions of standard phantoms of Asian man [9] and ethnicity-specific computational phantoms of a Korean man [10], Chinese man [5], Taiwanese man [11], and Brazilian man [12] are also available.

The basic approach for creating modern phantoms involves obtaining successive CT and micro -CT images of sections of cadavers [6,7,10]. The resulting image-based models can be scaled to the masses of reference individuals [2,13]. Currently, a number of research organizations are creating collections of phantoms [14,15] obtained by scanning bodies and their fragments, which makes it possible to increase the representativeness of computational phantoms and assess the impact of individual variability of anatomical features on the dose estimation uncertainty.

Micron structures of skeletal target tissues (e.g., active red bone marrow; AM) are not typically modeled in phantoms describing the whole body. This is due to the complex spongiosa microarchitecture, which requires high resolution of ≤ 200 μm [6]. As a result, skeletal models are being developed separately for AM dosimetry. The most advanced models are constructed by combining micro and macro images of bones [16]. The main problem with this approach is that the spongiosa voxel model is obtained based on fragments of autopsy material from different bones of a cadaver. This approach assumes that the bone microstructures of available samples correspond to those expected of reference individuals. However, information on the

reference microarchitecture of human bones in different parts of the skeleton is not yet available [8,17]. Therefore, it is impossible to assess the adequacy of the autopsy material used and to scale existing models.

Another problem is a biased estimation of cortical thickness values, because they are less than the typical CT resolution and cannot be obtained from images. In ICRP phantoms the thickness of cortical layer around the spongiosa is assigned to be equal to one voxel layer; the cortical bone at the long bones' shafts is thicker, and its thickness was adjusted to fit the reference value [18]. In other words, the description of the cortical layer is somewhat arbitrary and may differ significantly from reality.

To calculate the doses more correctly, it is necessary to find ways to avoid the problems described above, especially with regard to the doses in AM exposed to beta-emitters (such as $^{89,90}$Sr). Indeed, the mean free path length of $^{90}$Sr + $^{90}$Y electrons in bone tissue is about 1.3 mm, which is comparable to the cortical thickness (*Ct.Th*). Therefore, the accurate consideration of *Ct.Th* is a high priority. Another important parameter is the fraction of the trabecular bone volume in the spongiosa (*BV/TV*) which depends on the microarchitecture peculiarities and determines both the density of the spongiosa and the fraction of energy absorbed in AM. The factors influencing dose estimation in AM due to Sr isotopes in the cortical and trabecular bones are described in [19–22].

Thus, the creation of morphometrically adequate computational skeletal phantoms of people of different sex and ages is necessary to improve AM dosimetry for incorporated beta emitters. Unbiased dose estimates are critical for radiation risk evaluations in epidemiological studies of exposed populations. Therefore, the computational phantoms will be useful in terms of radiological protection. In particular, the importance of these issues is emphasized, for example, in the works of Kobayashi [23,24]. Our research group is engaged in reconstructing radiation doses to the population of the Southern Urals, where several industrial accidents during the middle of the previous century resulted in radioactive contamination of the environment including strontium isotopes [21,25–27]. We have developed a rod-like stochastic model of spongiosa [28] and a computational tool to generate phantoms [29]. The modeling introduces an error of < 12% to dose estimates that is 2–3 times lower than the uncertainty due to individual variability of morphometric parameters [21,22]. We elaborated the stochastic parametric skeletal dosimetry (SPSD) method [30] and analyzed hundreds of papers on skeleton anatomy and morphometry to define bone macro- and microarchitecture parameters [31,32]. Because the Urals population is represented mainly by Tatars, Bashkirs (Asian) and Russians (Caucasian), we combined available published data on skeletal morphology of Asians and Caucasians. In this paper we present the set of SPSD phantoms appropriate for the Urals population: newborns, 1-year, 5- year, 10- year, 15-year-old children (male and female), and adults (male and female).

## Materials and methods

The model was developed to calculate the radiation doses to hematopoetically active marrow (target tissue) resulting from $^{89,90}$Sr deposited in the trabecular and cortical bone (source tissues). Other basic principles of SPSD modeling were described in [30]. A specific feature of our approach is that the objects of modeling focus on skeletal regions with active hematopoiesis (hematopoietic sites), not the entire skeleton. The presence of active hematopoiesis in selected areas of the skeleton was assessed using Magnetic Resonance Imaging (MRI) data [33–43] to consider the degree of bone ossification. For example, the sternum and vertebral processes undergo significant ossification after the age of 5 and were not considered for younger ages. In contrast, the bones of the hands and feet contain active marrow (AM) only in the first year of life and have not been modeled for older ages. Hematopoietic sites are divided into segments with a relatively uniform spongiosa structure, uniform cortical thickness, and a shape approximated by simple geometric figures [44]. Segmentation makes it possible to generate phantoms in the voxel resolution required for adequate modeling of bone microstructure (from 40 μm).

Thus, each computational skeletal phantom is a model consisting of segments, each of which is described by a separate set of parameters. Parameters characterizing the segment dimensions can be divided into macro and micro

parameters. Macro-parameters include linear dimensions of segment models and *Ct.Th*. A cortical layer of a phantom segment was not assigned to all surfaces of a phantom. The cortex itself is assumed to be a homogeneous and isotropic bone substance. Spongiosa is represented by rod-like trabeculae of varying thickness, stochastically oriented in space and surrounded by AM. The microarchitecture of spongiosa is described by the following parameters: trabecular thickness – *Tb.Th*, trabecular separation – *Tb.Sp*, bone volume fraction of spongiosa – *BV/TV*, as well as the corresponding intra-specimen variabilities ($\sigma_{Tb.Th}$ and $\sigma_{Tb.Sp}$). The listed parameters are sufficient to generate a computational phantom of a bone segment using the "Trabecula" computer software [29]. To build a set of phantoms that simulate individual variability, an inter-specimen variability value corresponding to each parameter and a range of possible *BV/TV* values were needed [22].

The elemental composition and density of bone tissue and bone marrow are used as parameters in radiation transport simulation. Estimation of the skeleton-average energy absorbed in AM also requires the distribution of AM within the human skeleton of different ages. Thus, the parameters of computational phantoms can be divided into specific parameters of individual segments and general parameters for the entire skeleton.

## Segmentation

Bones of complex shape were divided into simpler segments. Bones and bone parts of regular shape, such as ribs, mid-sections of long bones, flat bones of the skull and ilium, were truncated so that all linear dimensions of the segment did not exceed 30 mm, i.e., only a fragment was modeled to simplify the Monte Carlo simulation. As shown in [20], $^{90}Sr + ^{90}Y$ electron emissions from adjacent bone segments have little effect on dose in AM due to their limited mean free path length in bone tissue and can be accounted for by a special correction in the dose calculation. A description of the geometric shapes of segment models, parameters characterizing their linear dimensions, and their designations are summarized in Table 1. See Tables 7–11 for bone sites' designated shapes.

Considering age-related changes in human anatomy, the same hematopoietic site can be segmented differently for different reference ages. An example of age-dependent segmentation of pelvic bones is shown in Fig 1.

## Parameterization of segment models

Only published data on bone size, shape, thickness of the cortical layer, and microarchitecture of bone tissue were analyzed for each skeletal segment [31,32]. In total, more than 400 publications were reviewed, including original scientific papers, and anatomical and histological atlases. The parameters of the spongiosa microarchitecture were evaluated using data from all available publications. For other bone parameters, only published data reported on individuals the authors

**Table 1. Description of the geometric shapes used to describe the segments, the corresponding parameters, and their designations.**

| Shape | Designation | Comment | Parameters |
|---|---|---|---|
| Box | b | Rectangular parallelepiped | $h \times a \times b$ (height × length × width) |
| Cylinder | c | Can have either circular or elliptic base | $h \times a \times b$ (height × major axis × minor axis) |
| Deformed cylinder | dc | A figure of irregular shape with two parallel elliptical bases and ruled lateral surface. The planes on which parallel axes lie are perpendicular to each other and the planes of the bases. A truncated cone is a particular case of the deformed cylinder. | $h \times a \times b \times c \times d$ (height × major axis of lower base × minor axis of lower base × major axis of upper base × minor axis of upper base, with axes of upper and lower bases being parallel |
| Triangular prism | p | Prism with isosceles triangular base is only used to describe a segment of pelvic *ischium acetabulum* and a segment of *sacral ala* | $h \times a \times b$ (height × triangular base length × triangular equal side length) |
| Ellipsoid | e | Typical of talus and calcaneus of both hand and foot; considered as hematopoietic for newborn only | $h \times a \times b$ (three axes of ellipsoid) |
| Tube | t | A figure bounded by the two nested circular cylinders; typical of pelvic acetabulum only | $h \times a \times b$ (height × diameter of outer cylinder × diameter of inner cylinder) |

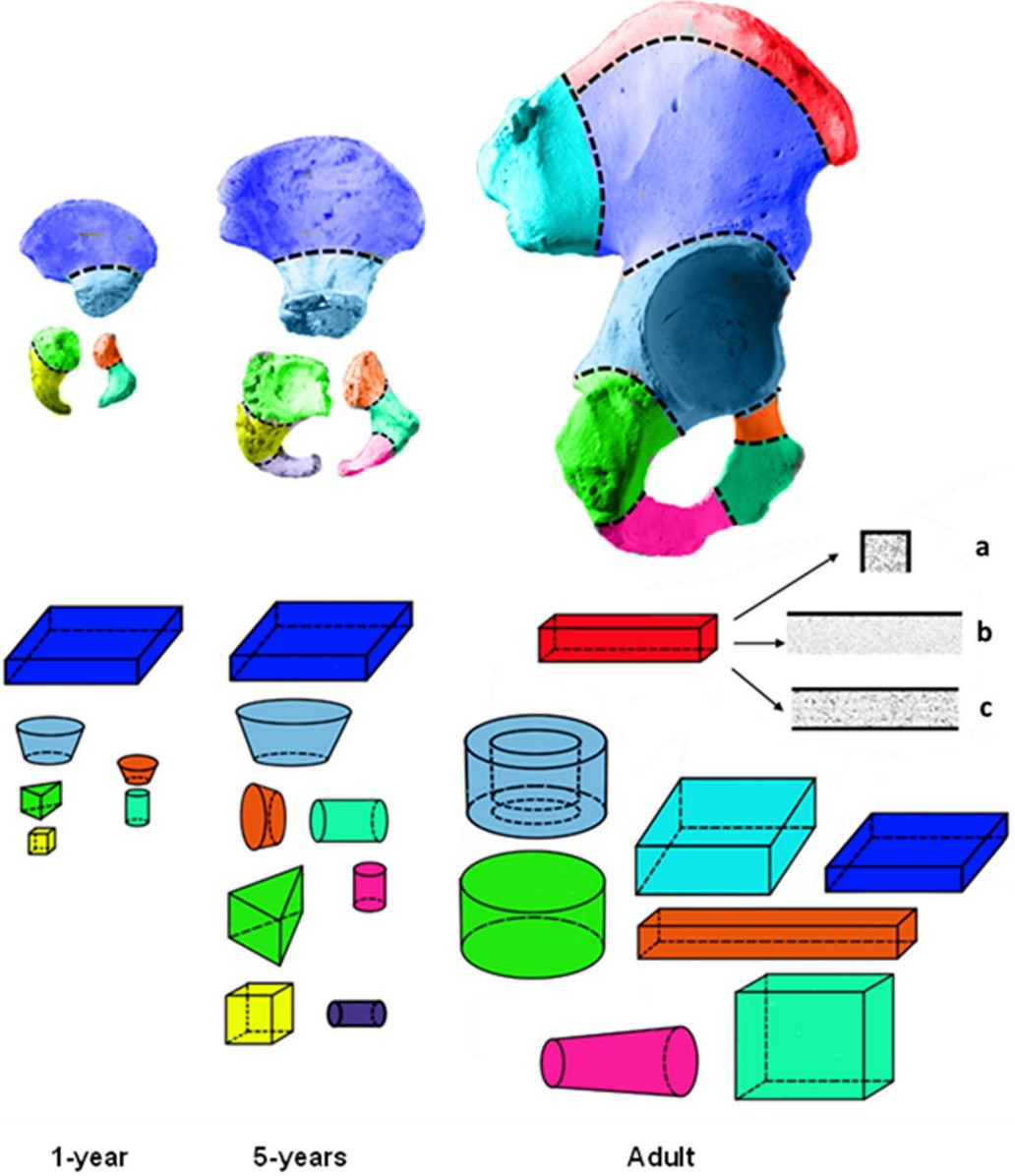

**Fig 1. Segmentation of human pelvic bones when creating age-specific phantoms (simulated bones are shown above, geometric approximations for bone segments are shown below).** For the iliac crest of adults (in red), visualization of phantom cross-sections is shown: a-frontal, b-sagittal, c-horizontal projections (the black solid lines mark the cortical bone, which does not cover all sides of spongiosa).

understood to be "healthy" were considered. Linear bone dimensions were estimated based only on published data for individuals evaluated in the second half of the 20th to the early 21st century (Caucasian and Asian). The greatest amount of published data on which the phantom parameters were based has been found for adults. For newborns, the amount of published data is more limited.

Details regarding the morphometric basis for estimating the parameters of computational SPSD phantoms is presented in [31,32].

The measurement data of standard characteristics used in anthropometry and practical medicine to describe the size and microarchitecture of various bones were averaged. As a result, for each skeletal segment with active hematopoiesis, a set of estimates of the population-average linear dimensions and characteristics of bone microstructure, as well as $\sigma_{Tb.Th}$ and $\sigma_{Tb.Sp}$ were obtained. These values were taken as basic parameters of phantom segments. Additionally, individual variabilities of bone segment dimensions were assessed to create not only reference (basic) phantoms, but also supplementary phantoms.

### Segment phantoms

For each segment, a voxel phantom was generated using segment-specific parameters with the Trabecula program [29]. Voxel resolution was set to $0.7 \times Tb.Th$. This resolution does not lead to systematic error in calculation of dose rates in AM due to $^{90}Sr/^{90}Y$ and $^{89}Sr$ in bone [22]. Each voxel simulated bone marrow or mineralized bone tissue in the trabeculae or cortical layer. The volumes of the simulated media were automatically calculated in the Trabecula program as the sum of the volumes of the corresponding voxels. If the segment corresponded to a truncated bone fragment, then the resulting media volumes were recalculated for the entire bone. Fig 2 shows a comparison of cross sections of a real image of the femoral neck of an adult and a phantom of the same bone segment.

### Assessment of parameters characteristic of the skeleton as a whole

In addition to the parameters specific to individual segments, there is also a set of parameters specific to the skeleton of a certain age as a whole. ICRP data on elemental composition of bone tissue and bone marrow were used in the study [8]. It has previously been shown that the elemental composition does not significantly affect dose estimates [21], unlike bone density, which can change significantly over time. The age dependence of bone density was estimated based on published data [45–49].

## Results

### Parameters characteristic of the skeleton as a whole

In newborns, AM is found in almost all the areas of the skeleton that have undergone ossification. With age, hematopoiesis stops in the bones of the hands and feet, as well as in the distal tubular bones. In adults, hematopoiesis is absent in the bones of the upper and lower limbs, except for the proximal humerus and femur. Other areas of the skeleton also undergo age-related changes. For example, hematopoiesis in the body of the scapula (flat part) terminates at the age of 1–3 [39]. Therefore, this segment was modeled in a phantom representing a child aged 0–1. The sternum, as previously mentioned, was modeled for reference ages ≥ 5 years because it is formed mainly by cartilaginous tissue up to the age of 5. Thus, the number of hematopoietic sites changes with age. AM fractions in each site are parameters of the skeletal phantom with estimates provided in Table 2. Skeletal sites with active hematopoiesis for reference ages are illustrated in Fig 3.

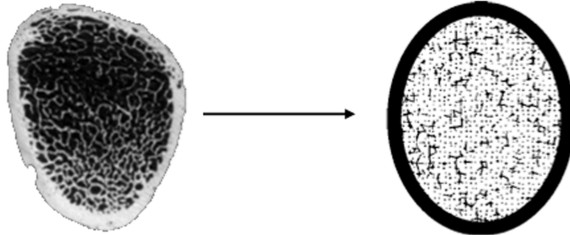

**Fig 2. A typical section of the femoral neck (left) and its corresponding phantom (right) (the trabeculae and cortical bone are shown in black, the AM is shown in white).**

**Table 2. Active marrow distribution (%) considered in the computational skeleton phantoms of different age (based on [50,51]).**

| Site | Phantoms | | | | | |
|---|---|---|---|---|---|---|
| | Newborn | 1-Y | 5-Y | 10-Y | 15-Y ♂,♀ | Adults ♂,♀ |
| Femur | 6.7 | 8.1 | 13.5 | 15.7 | 11.3 | 5.95 |
| Humerus | 4.5 | 5.2 | 4.8 | 4.1 | 3.8 | 3.6 |
| Sacrum | 4.4 | 5.1 | 5.7 | 6.8 | 8.5 | 7.48 |
| Tibia and Fibula | 7.1 | 8.7 | 9.3 | 5.6 | – † | – |
| Pelvis | 11.4 | 13.1 | 13.5 | 15.8 | 18.6 | 23.2 |
| Skull | 28.2 | 28.7 | 18.1 | 12.8 | 10.2 | 6.2 |
| Clavicle | 0.7 | 0.9 | 0.9 | 0.9 | 1.0 | 15.3 |
| Scapula | 2.3 | 2.7 | 2.8 | 2.9 | 3.3 | |
| Ribs | 7.1 | 8.2 | 9.1 | 11.0 | 13.7 | |
| Radius and ulnae | 2.4 | 2.6 | 2.1 | – | – | – |
| Hand and foot | 10.8 | – | – | – | – | – |
| Sternum | – | – | 1.8 | 2.1 | 1.8 | 1.8 |
| Cervical vertebrae | 1.7 | 2.1 | 2.3 | 2.7 | 3.3 | 3.5 |
| Thoracic vertebrae | 7.2 | 8.3 | 9.2 | 11.0 | 13.8 | 17.5 |
| Lumbar vertebrae | 5.5 | 6.4 | 7.0 | 8.5 | 10.6 | 15.5 |

† a "–" symbol indicates there is no AM for this bone site and age group.

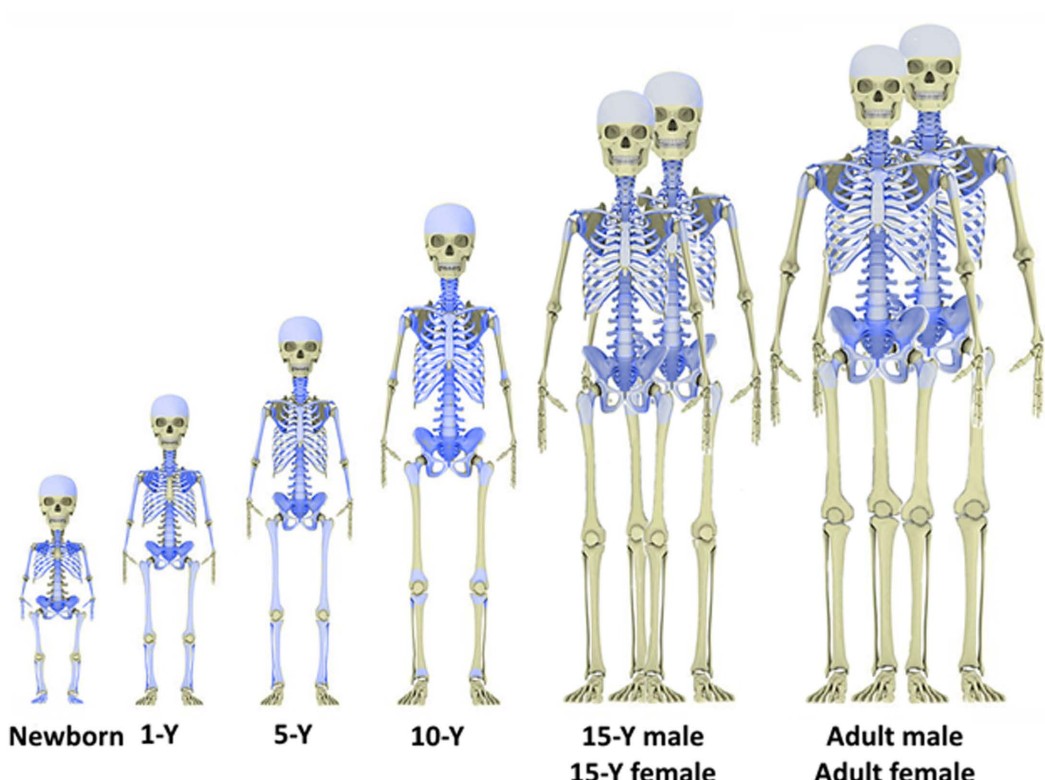

**Fig 3. Skeletal areas with active hematopoiesis (highlighted in blue) for different reference ages.**

The distribution of AM within each segment of the listed sites was assumed to be uniform. Uncertainty in the estimation of the parameters in Table 2 is associated with individual variability. For adults, the average uncertainty is 35% (according to [50]), and for other reference ages it makes up 40% [22].

The elemental composition (common/shared parameters for all reference ages) of bone tissue and bone marrow are presented in Table 3.

Bone density increases with age from 1.65 g/cm$^3$ for newborns to 1.9 g/cm$^3$ for adults [45–49]. Age-related changes in bone density are demonstrated in Fig 4.

The densities of the simulated media used as parameters of the computational phantoms are presented in Table 4. Bone marrow density for children was assumed to be equal to the density of water – 1 g/cm$^3$; for adults, bone marrow density was assumed to be equal to that of adipose tissue – 0.98 g/cm$^3$ [8].

**Table 3. Mass fraction of elements in simulated media (according to [8]).**

| Mass No. | Element | Mass fraction, % | |
|---|---|---|---|
| | | Bone | Bone marrow |
| 1 | H | 3.50 | 10.50 |
| 12 | C | 16.00 | 41.40 |
| 14 | N | 4.20 | 3.40 |
| 16 | O | 44.50 | 43.90 |
| 23 | Na | 0.30 | 0.10 |
| 24 | Mg | 0.20 | 0.20 |
| 31 | P | 9.50 | 0.20 |
| 32 | S | 0.30 | 0.20 |
| 40 | Ca | 21.50 | – |

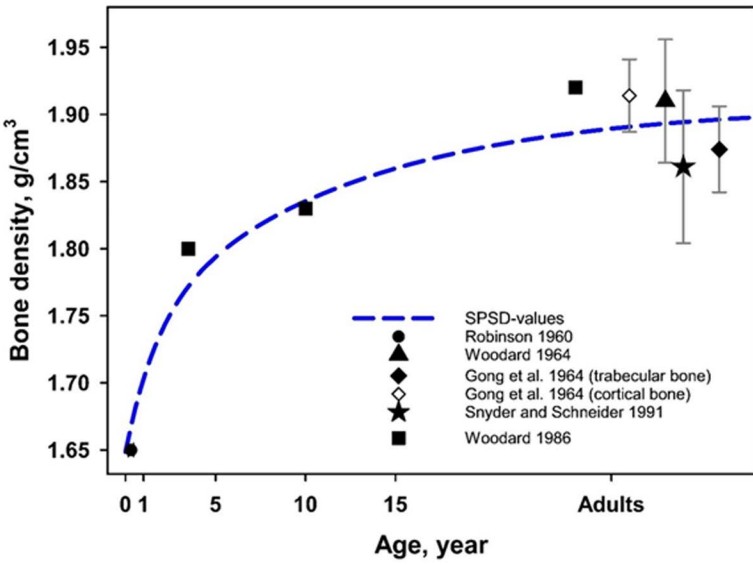

**Fig 4. Age-related changes in the density of bone tissue.**

**Table 4. The densities of the SPSD simulated media (g/cm³).**

| Media | Phantom | | | | | |
|---|---|---|---|---|---|---|
| | Newborn | 1-Y | 5-Y | 10-Y | 15-Y♂,♀ | Adults♂,♀ |
| Bone tissue | 1.65 | 1.70 | 1.80 | 1.83 | 1.85 | 1.90 |
| Bone marrow | 1.0 | | | | | 0.98 |

## Segment-specific parameters of SPSD phantoms

The number and shape of segments within the same skeleton site may change with age. For example, age-related changes are evident in the example of pelvic bones in Fig 1. In particular, the shape of the ischial bone (highlighted in bright green) in children is well approximated by a prism with a triangular base, and in adults by a cylinder. In total, 289 models of the unique segments were created to describe the family of reference phantoms. The number of segments and their distribution by shape are shown in Table 5.

As shown in Table 5, 77% of the skeleton segments are stylized with boxes and cylinders. Table 6 presents the volumes of trabecular bone (TBV), cortical bone (CBV) and bone marrow (BMV) typical of reference SPSD phantoms (within hematopoietic fragments of skeleton) as well as their coefficients of variation (CVs) due to morphometric individual variability. The parameters of the segments of the reference phantoms are presented in Tables 7–14. These are very large tables containing all the necessary parameters for generating bone segment phantoms.

**Table 5. The number of segments and their distribution by shape for different reference ages.**

| Reference age | Number of segments | Number of segments of given shape† | | | | | |
|---|---|---|---|---|---|---|---|
| | | b | c | dc | p | e | t |
| Newborn | 34 | 11 | 13 | 9 | 0 | 1 | 0 |
| 1-Y | 39 | 15 | 13 | 10 | 1 | 0 | 0 |
| 5-Y | 43 | 17 | 15 | 10 | 1 | 0 | 0 |
| 10-Y | 38 | 17 | 11 | 9 | 1 | 0 | 0 |
| 15-Y ♂, ♀ | 28‡ + 18♂ + 18♀ | 39 | 14 | 9 | 1 | 0 | 1 |
| Adults ♂, ♀ | 23‡ + 24♂ + 24♀ | 43 | 17 | 9 | 1 | 0 | 1 |
| **Total** | **289** | 142 | 83 | 56 | 5 | 1 | 2 |

† b = box, c = cylinder, dc = deformed cylinder, p = triangular prism, e = ellipsoid, t = tube.

‡ the segments are not sex-specific.

**Table 6. Volumetric characteristics of SPSD phantoms including reference values (cm³) and corresponding individual variability in terms of CV, %.**

| Phantom | | TBV | | CBV | | BMV | |
|---|---|---|---|---|---|---|---|
| | | Total | CV | Total | CV | Total | CV |
| Newborn | | 40.7 | 29 | 31.9 | 26 | 72.8 | 39 |
| 1-Y | | 67.6 | 57 | 117.3 | 20 | 183.6 | 50 |
| 5-Y | | 173.4 | 39 | 314.5 | 18 | 547.6 | 40 |
| 10-Y | | 258.8 | 34 | 340.7 | 14 | 862.0 | 35 |
| 15-Y | ♂ | 321.5 | 35 | 495.1 | 19 | 1332.0 | 31 |
| | ♀ | 282.6 | 29 | 425.4 | 18 | 1130.8 | 30 |
| Adults | ♂ | 344.6 | 34 | 590.0 | 19 | 1674.7 | 29 |
| | ♀ | 297.2 | 33 | 511.0 | 19 | 1420.1 | 32 |

**Table 7. Parameters of bone segment phantoms for a reference newborn; macro-architecture (*h, a, b, c, d, Ct.Th*) and microarchitecture (*Tb. Th*, *Tb.Sp*) values are given in mm; *BV/TV* is in relative units; variability ($\sigma_{Tb.Th}$ and $\sigma_{Tb.Sp}$) is given in percentages; the shape of a segment is designated as follows: c-cylinder, dc-deformed cylinder, b-box, e-ellipsoid.**

| Site | Segment | Shape | h | a | b | c | d | Ct.Th | Face, covered by Ct.Th | BV/TV | Tb.Th | σTb.Th | Tb.Sp | σTb.Sp |
|---|---|---|---|---|---|---|---|---|---|---|---|---|---|---|
| Clavicle | Acromeon end | dc | 5.9 | 10 | 5.9 | 5.9 | 4.3 | 0.3 | lateral | 0.29 | 0.15 | 22 | 0.8 | 23 |
| Clavicle | Shaft | c | 33 | 5.9 | 4.3 | | | 0.8 | lateral | 0.29 | 0.15 | 22 | 0.8 | 23 |
| Clavicle | Sternal end | dc | 5.9 | 12 | 10 | 5.9 | 4.3 | 0.3 | lateral | 0.29 | 0.15 | 22 | 0.8 | 23 |
| Femur | Distal end | dc | 19 | 26 | 12 | 7.2 | 7.2 | 0.4 | lateral | 0.37 | 0.11 | 22 | 0.39 | 23 |
| Femur | Proximal end | dc | 19 | 26 | 12 | 7.2 | 7.2 | 0.5 | lateral | 0.37 | 0.11 | 22 | 0.39 | 23 |
| Femur | Shaft | c | 30 | 7.2 | 7.2 | | | 1.7 | lateral | 0.37 | 0.11 | 22 | 0.39 | 23 |
| Hand and foot | Precarpal | e | | 7.8 | 12 | 7.8 | | 0.2 | total | 0.22 | 0.12 | 22 | 0.25 | 23 |
| Hand and foot | Tube bones | c | 8.9 | 3.8 | 3.8 | | | 0.2 | lateral | 0.22 | 0.12 | 22 | 0.248 | 23 |
| Humerus | Distal end | dc | 13 | 17 | 6 | 6 | 6 | 0.3 | lateral | 0.28 | 0.102 | 22 | 0.36 | 23 |
| Humerus | Proximal end | dc | 13 | 13 | 13 | 6 | 6 | 0.4 | lateral | 0.28 | 0.102 | 22 | 0.36 | 23 |
| Humerus | Shaft | c | 30 | 6 | 6 | | | 1.3 | lateral | 0.28 | 0.102 | 22 | 0.36 | 23 |
| Pelvis | Ilium part 1 | b | 4 | 24 | 24 | | | 1.2[1] 0.5[1] | ab1 ab2 | 0.32 | 0.166 | 10 | 0.32 | 10 |
| Pelvis | Ilium part 2 | b | 4 | 20 | 20 | | | 0.2 | ab | 0.31 | 0.166 | 10 | 0.32 | 10 |
| Pelvis | Ischium | c | 7.5 | 18 | 12 | | | 0.4 | ab1 | 0.31 | 0.166 | 10 | 0.32 | 10 |
| Pelvis | Pubis | c | 16 | 7.5 | 7.5 | | | 0.4 | lateral | 0.31 | 0.166 | 10 | 0.32 | 10 |
| Radius and ulna | End | dc | 12 | 5.8 | 5.8 | 3.9 | 3.9 | 0.3 | lateral | 0.21 | 0.08 | 22 | 0.51 | 23 |
| Radius and ulna | Shaft | c | 30 | 3.9 | 3.9 | | | 0.9 | lateral | 0.21 | 0.08 | 22 | 0.51 | 23 |
| Ribs | Ribs | b | 5.7 | 30 | 3.2 | | | 0.4 | ha; ab | 0.19 | 0.136 | 22 | 0.52 | 23 |
| Sacrum | Body 1 | b | 6.3 | 15 | 7.5 | | | | | 0.45 | 0.096 | 48 | 0.6 | 43 |
| Sacrum | Body 2 | b | 6.3 | 12 | 6 | | | | | 0.45 | 0.096 | 48 | 0.6 | 43 |
| Sacrum | Body 3 | b | 5.7 | 8.9 | 5.3 | | | | | 0.45 | 0.096 | 48 | 0.6 | 43 |
| Sacrum | Body 4 | b | 3.8 | 8.9 | 5.3 | | | | | 0.45 | 0.096 | 48 | 0.6 | 43 |
| Sacrum | Body 5 | b | 3.8 | 7.5 | 3.8 | | | | | 0.45 | 0.096 | 48 | 0.6 | 43 |
| Scapula | Acromion | b | 7 | 16 | 13 | | | 0.4 | ha; hb1; ab | 0.28 | 0.12 | 22 | 0.48 | 23 |
| Scapula | Body | b | 2.7 | 30 | 30 | | | 0.4 | ab | 0.28 | 0.12 | 22 | 0.48 | 23 |
| Scapula | Glenoid | c | 5.4 | 10 | 7.6 | | | 0.5 | lateral | 0.28 | 0.12 | 22 | 0.48 | 23 |
| Skull | Flat bones | b | 2 | 30 | 30 | | | | | 0.53 | 0.29 | 8 | 0.57 | 15 |
| Tibia and fibula | Distal end | dc | 15 | 15 | 15 | 6.9 | 6.9 | 0.3 | lateral | 0.35 | 0.075 | 22 | 0.49 | 23 |
| Tibia and fibula | Fibula shaft | c | 30 | 2.9 | 2.9 | | | 0.7 | lateral | 0.34 | 0.075 | 22 | 0.49 | 23 |
| Tibia and fibula | Proximal end | dc | 19 | 21 | 13 | 6.9 | 6.9 | 0.3 | lateral | 0.35 | 0.075 | 22 | 0.49 | 23 |
| Tibia and fibula | Tibia shaft | c | 30 | 6.9 | 6.9 | | | 1.4 | lateral | 0.35 | 0.075 | 22 | 0.49 | 23 |
| Vertebra | C-body | c | 4.1 | 6.9 | 6.5 | | | | | 0.6 | 0.25 | 48 | 0.6 | 43 |
| Vertebra | L-body | c | 7.1 | 15 | 7.7 | | | | | 0.45 | 0.096 | 48 | 0.6 | 43 |
| Vertebra | T-body | c | 5.1 | 11 | 7.6 | | | | | 0.45 | 0.096 | 48 | 0.6 | 43 |

The minimum linear dimension was the height of the box corresponding to the flat bones of a newborn's skull, h = 0.2 cm (Table 7); the maximum value corresponded to the length of the box of the 1st segment of a 10-Y sacrum, a = 8.8 cm (Table 10). The minimum BV/TV value was recorded in the adult proximal humerus phantom (0.06) and the maximum ratio – in the neonatal cervical vertebral body phantom (0.6). Cortical thickness is in the range 0.01 (sternum of 5-Y) to 0.37 (femur shaft of 5-Y) cm. The *Ct.Th* parameter in Tables 7–14 requires separate comments. Not all surfaces of segment phantoms can be covered with a cortical layer, and some segments have different cortical thicknesses on different surfaces (See Fig 1).

**Table 8. Parameters of bone segment phantoms for a reference 1-year-old; macro-architecture (*h, a, b, c, d, Ct.Th*) and microarchitecture (*Tb. Th, Tb.Sp*) values are given in mm; *BV/TV* is in relative units; variability ($\sigma_{Tb.Th}$ and $\sigma_{Tb.Sp}$) is given in percentages; the shape of a segment is designated as follows: c-cylinder, dc-deformed cylinder, b-box, p-prism with a triangle base.**

| Site | Segment | Shape | h | a | b | c | d | Ct.Th | Face, covered by Ct.Th | BV/TV | Tb.Th | σTb.Th | Tb.Sp | σTb.Sp |
|---|---|---|---|---|---|---|---|---|---|---|---|---|---|---|
| Clavicle | Acromial end | dc | 7.4 | 12 | 7.2 | 7.2 | 5.2 | 0.4 | lateral | 0.29 | 0.15 | 22 | 0.8 | 23 |
| Clavicle | Shaft | c | 42 | 7.2 | 5.2 | | | 0.9 | lateral | 0.29 | 0.15 | 8 | 0.8 | 15 |
| Clavicle | Sternal end | dc | 7.4 | 14 | 13 | 7.2 | 5.2 | 0.4 | lateral | 0.29 | 0.15 | 22 | 0.8 | 23 |
| Femur | Distal end | dc | 36 | 34 | 18 | 11 | 11 | 0.6 | lateral | 0.22 | 0.16 | 22 | 0.54 | 23 |
| Femur | Proximal end | dc | 36 | 34 | 18 | 11 | 11 | 0.7 | lateral | 0.22 | 0.16 | 22 | 0.54 | 23 |
| Femur | Shaft | c | 30 | 11 | 11 | | | 2.3 | lateral | 0.22 | 0.16 | 22 | 0.54 | 23 |
| Humerus | Distal end | dc | 16 | 20 | 9.1 | 9.1 | 9.1 | 0.4 | lateral | 0.22 | 0.174 | 22 | 0.58 | 23 |
| Humerus | Proximal end | dc | 16 | 20 | 20 | 9.1 | 9.1 | 0.5 | lateral | 0.22 | 0.174 | 22 | 0.58 | 23 |
| Humerus | Shaft | c | 30 | 9.1 | 9.1 | 9.1 | | 1.6 | lateral | 0.22 | 0.174 | 22 | 0.58 | 23 |
| Pelvis | Ilium acetabular part | c | 15 | 26 | 10 | 24 | 18 | 0.4 | lateral | 0.23 | 0.123 | 10 | 0.48 | 10 |
| Pelvis | Ilium flat part 1 | b | 5 | 30 | 30 | | | 1.2[1] ab1<br>0.5[1] ab2 | | 0.23 | 0.123 | 10 | 0.48 | 10 |
| Pelvis | Ilium flat part 2 | b | 5 | 30 | 30 | | | 0.4 | ab | 0.23 | 0.123 | 10 | 0.48 | 10 |
| Pelvis | Ischium acetabular part | p | 18 | 18 | 18 | | | 0.4 | ah; bh; ch | 0.23 | 0.123 | 10 | 0.48 | 10 |
| Pelvis | Ischium tuberosity | b | 13 | 12 | 12 | | | 0.4 | ha; hb | 0.23 | 0.123 | 10 | 0.48 | 10 |
| Pelvis | Pubic ramus superior | c | 19 | 7.7 | 7.7 | | | 0.4 | lateral | 0.23 | 0.123 | 10 | 0.48 | 10 |
| Pelvis | Pubis acetabular part | dc | 4.8 | 16 | 11 | 7.7 | 7.7 | 0.4 | lateral | 0.23 | 0.123 | 10 | 0.48 | 10 |
| Radius and ulna | End | dc | 16 | 8 | 5.3 | 5.3 | 5.3 | 0.4 | lateral | 0.16 | 0.134 | 22 | 0.77 | 23 |
| Radius and ulna | Shaft | c | 30 | 5.3 | 5.3 | | | 1.1 | lateral | 0.16 | 0.134 | 22 | 0.77 | 23 |
| Ribs | Ribs | b | 8.7 | 30 | 3.9 | | | 0.5 | ha; ab | 0.29 | 0.231 | 22 | 0.51 | 23 |
| Sacrum | Ala 1 | b | 9.2 | 11 | 13 | | | | | 0.14 | 0.096 | 48 | 0.6 | 43 |
| Sacrum | Ala 2 | b | 9.2 | 8 | 10 | | | | | 0.14 | 0.096 | 48 | 0.6 | 43 |
| Sacrum | Ala 3 | b | 8.3 | 8 | 8.8 | | | | | 0.14 | 0.096 | 48 | 0.6 | 43 |
| Sacrum | Ala 4 | b | 5.5 | 5.4 | 8.8 | | | | | 0.14 | 0.096 | 48 | 0.6 | 43 |
| Sacrum | Body | b | 9.2 | 25 | 13 | | | | | 0.14 | 0.096 | 48 | 0.6 | 43 |
| Sacrum | Body | b | 9.2 | 20 | 10 | | | | | 0.14 | 0.096 | 48 | 0.6 | 43 |
| Sacrum | Body | b | 8.3 | 15 | 8.8 | | | | | 0.14 | 0.096 | 48 | 0.6 | 43 |
| Sacrum | Body | b | 5.5 | 15 | 8.8 | | | | | 0.14 | 0.096 | 48 | 0.6 | 43 |
| Sacrum | Body | b | 5.5 | 13 | 5 | | | | | 0.14 | 0.096 | 48 | 0.6 | 43 |
| Scapula | Acromion | b | 7 | 16 | 13 | | | 0.4 | ha; hb1; ab | 0.22 | 0.192 | 22 | 0.96 | 23 |
| Scapula | Body | b | 2.7 | 30 | 30 | | | 0.4 | ab | 0.22 | 0.192 | 22 | 0.96 | 23 |
| Scapula | Glenoid | c | 6.8 | 18 | 10 | | | 0.5 | lateral | 0.22 | 0.192 | 22 | 0.96 | 23 |
| Skull | Flat bones | b | 2.7 | 30 | 30 | | | 0.7 | ab | 0.52 | 0.29 | 10 | 0.57 | 10 |
| Tibia and fibula | Distal end | dc | 22 | 17 | 17 | 9 | 9 | 0.5 | lateral | 0.2 | 0.089 | 22 | 0.74 | 23 |
| Tibia and fibula | Fibula shaft | c | 30 | 4.4 | 4.4 | | | 1.2 | lateral | 0.2 | 0.089 | 22 | 0.74 | 23 |
| Tibia and fibula | Proximal end | dc | 39 | 27 | 15 | 9 | 9 | 0.5 | lateral | 0.2 | 0.089 | 22 | 0.74 | 23 |
| Tibia and fibula | Tibia shaft | c | 30 | 9 | 9 | | | 2.3 | lateral | 0.2 | 0.089 | 22 | 0.74 | 23 |
| Vertebra | Cervical vertebra body | c | 5.8 | 13 | 9.7 | | | | | 0.2 | 0.18 | 48 | 0.6 | 43 |
| Vertebra | Lumbar vertebra body | c | 9.6 | 21 | 9.6 | | | | | 0.14 | 0.096 | 48 | 0.6 | 43 |
| Vertebra | Thoracic vertebra body | c | 8 | 15 | 12 | | | | | 0.14 | 0.096 | 48 | 0.6 | 43 |

**Table 9. Parameters of bone segment phantoms for a reference 5-year-old; macro-architecture (*h, a, b, c, d, Ct.Th*) and microarchitecture (*Tb.Th, Tb.Sp*) values are given in mm; *BV/TV* is in relative units; variability ($\sigma_{Tb.Th}$ and $\sigma_{Tb.Sp}$) is given in percentages; the shape of a segment is designated as follows: c-cylinder, dc-deformed cylinder, b-box, p-prism with a triangle base.**

| Site | Segment | Shape | h | a | b | c | d | Ct.Th | Face, covered by Ct.Th | BV/TV | Tb.Th | σTb.Th | Tb.Sp | σTb.Sp |
|---|---|---|---|---|---|---|---|---|---|---|---|---|---|---|
| Clavicle | Acromial end | dc | 12 | 15 | 8.7 | 8.7 | 6.8 | 0.5 | lateral | 0.29 | 0.15 | 22 | 0.8 | 23 |
| Clavicle | Shaft | c | 30 | 8.7 | 6.8 | | | 1.1 | lateral | 0.15 | 0.197 | 22 | 0.8 | 23 |
| Clavicle | Sternal end | dc | 12 | 18 | 16 | 8.7 | 6.8 | 0.5 | lateral | 0.29 | 0.15 | 22 | 0.8 | 23 |
| Femur | Distal end | dc | 49 | 68 | 25 | 17 | 17 | 1.1 | lateral | 0.26 | 0.24 | 22 | 0.54 | 23 |
| Femur | Lower proximal end | c | 25 | 23 | 23 | | | 1.3 | lateral | 0.35 | 0.24 | 22 | 0.54 | 23 |
| Femur | Shaft | c | 30 | 17 | 17 | | | 3.7 | lateral | 0.26 | 0.24 | 22 | 0.54 | 23 |
| Femur | Upper proximal end | c | 25 | 23 | 23 | | | 1.3 | lateral | 0.35 | 0.24 | 22 | 0.54 | 23 |
| Humerus | Distal end | dc | 27 | 46 | 15 | 15 | 15 | 0.8 | lateral | 0.22 | 0.208 | 22 | 0.58 | 23 |
| Humerus | Proximal end | dc | 20 | 32 | 32 | 15 | 15 | 0.9 | lateral | 0.22 | 0.208 | 22 | 0.58 | 23 |
| Humerus | Shaft | c | 30 | 15 | 15 | | | 2.5 | lateral | 0.22 | 0.208 | 22 | 0.58 | 23 |
| Pelvis | Ilium acetabular part | dc | 20 | 35 | 16 | 34 | 27 | 0.8 | lateral | 0.25 | 0.153 | 10 | 0.481 | 10 |
| Pelvis | Ilium flat part 1 | b | 7.9 | 30 | 30 | | | 1.6[1] 0.8[1] | ab1 ab2 | 0.25 | 0.153 | 10 | 0.481 | 10 |
| Pelvis | Ilium flat part 2 | b | 7.9 | 30 | 30 | | | 0.8 | ab | 0.25 | 0.153 | 10 | 0.481 | 10 |
| Pelvis | Ischial ramus inferior | c | 19 | 8.8 | 8.8 | | | 0.5 | h | 0.25 | 0.153 | 10 | 0.481 | 10 |
| Pelvis | Ischium acetabular part | p | 21 | 27 | 21 | | | 0.5 | ah; bh; ch | 0.25 | 0.153 | 10 | 0.481 | 10 |
| Pelvis | Ischium tuberosity | b | 25 | 18 | 14 | | | 0.5 | ha; hb | 0.25 | 0.153 | 10 | 0.481 | 10 |
| Pelvis | Pubis acetabular part | dc | 7.3 | 22 | 18 | 13 | 8.8 | 0.5 | lateral | 0.25 | 0.153 | 10 | 0.481 | 10 |
| Pelvis | Pubis ramus inferior | c | 19 | 8.8 | 8.8 | | | 0.5 | lateral | 0.25 | 0.153 | 10 | 0.481 | 10 |
| Pelvis | Pubis ramus superior | c | 29 | 13 | 8.8 | | | 0.5 | lateral | 0.25 | 0.153 | 10 | 0.481 | 10 |
| Radius and ulna | End | dc | 26 | 13 | 8.3 | 8.3 | 8.3 | 0.5 | lateral | 0.16 | 0.16 | 22 | 0.765 | 23 |
| Radius and ulna | Shaft | c | 30 | 8.3 | 8.3 | | | 1.5 | lateral | 0.16 | 0.16 | 22 | 0.765 | 23 |
| Ribs | Ribs | b | 9.4 | 30 | 4.4 | | | 0.5 | ha; ab | 0.2 | 0.231 | 22 | 0.505 | 23 |
| Sacrum | Body-ala 1 | b | 17 | 75 | 21 | | | 0.7 | ha | 0.14 | 0.096 | 48 | 0.6 | 43 |
| Sacrum | Body-ala 2 | b | 16 | 60 | 15 | | | 0.7 | ha | 0.14 | 0.096 | 48 | 0.6 | 43 |
| Sacrum | Body-ala 3 | b | 14 | 52 | 11 | | | 0.7 | ha | 0.14 | 0.096 | 48 | 0.6 | 43 |
| Sacrum | Body-ala 4 | b | 10 | 45 | 6.4 | | | 0.7 | ha | 0.14 | 0.096 | 48 | 0.6 | 43 |
| Sacrum | Body-ala 5 | b | 10 | 22 | 6.4 | | | 0.7 | ha | 0.14 | 0.096 | 48 | 0.6 | 43 |
| Scapula | Acromion | b | 7.6 | 20 | 16 | | | 0.8 | ha; hb1; ab | 0.22 | 0.24 | 22 | 0.964 | 23 |
| Scapula | Glenoid | c | 12 | 25 | 18 | | | 0.9 | lateral | 0.22 | 0.24 | 22 | 0.964 | 23 |
| Scapula | Lateral margin | b | 30 | 3.2 | 10 | | | 0.8 | ha1; hb | 0.22 | 0.24 | 22 | 0.964 | 23 |
| Skull | Flat bones | b | 4.2 | 30 | 30 | | | 1.1 | ab | 0.52 | 0.29 | 8 | 0.57 | 15 |
| Sternum | Sternum | b | 6.9 | 30 | 30 | | | 0.1 | ab | 0.15 | 0.135 | 22 | 1 | 23 |
| Tibia and fibula | Distal end | dc | 34 | 24 | 24 | 15 | 15 | 0.7 | lateral | 0.25 | 0.126 | 22 | 0.735 | 23 |
| Tibia and fibula | Fibula body | c | 30 | 8.1 | 8.1 | | | 1.5 | lateral | 0.25 | 0.126 | 22 | 0.735 | 23 |
| Tibia and fibula | Proximal end | dc | 34 | 55 | 22 | 15 | 15 | 0.7 | lateral | 0.25 | 0.126 | 22 | 0.735 | 23 |
| Tibia and fibula | Tibia shaft | c | 30 | 15 | 15 | | | 2.9 | lateral | 0.25 | 0.126 | 22 | 0.735 | 23 |
| Vertebra | C-body | c | 7.3 | 18 | 11 | | | 0.2 | lateral | 0.21 | 0.14 | 48 | 0.6 | 43 |
| Vertebra | L-body | c | 16 | 34 | 23 | | | 0.2 | lateral | 0.14 | 0.096 | 48 | 0.6 | 43 |
| Vertebra | L-spinous proc | b | 15 | 13 | 5 | | | 0.2 | ha; ab; hb1 | 0.14 | 0.096 | 48 | 0.6 | 43 |
| Vertebra | L-transverse proc | b | 6.4 | 12 | 5 | | | 0.2 | ha; ab; hb1 | 0.14 | 0.096 | 48 | 0.6 | 43 |
| Vertebra | T-body | c | 12 | 21 | 17 | | | 0.2 | lateral | 0.14 | 0.096 | 48 | 0.6 | 43 |
| Vertebra | T-spinous proc | b | 5.9 | 17 | 3 | | | 0.2 | ha; ab; hb1 | 0.14 | 0.096 | 48 | 0.6 | 43 |
| Vertebra | T-transverse proc | b | 7.3 | 11 | 5.3 | | | 0.2 | ha; ab; hb1 | 0.14 | 0.096 | 48 | 0.6 | 43 |

**Table 10. Parameters of bone segment phantoms for a reference 10-year-old; macro-architecture (*h, a, b, c, d, Ct.Th*) and microarchitecture (*Tb.Th*, *Tb.Sp*) values are given in mm; *BV/TV* is in relative units; variability ($\sigma_{Tb.Th}$ and $\sigma_{Tb.Sp}$) is given in percentages; the shape of a segment is designated as follows: c-cylinder, dc-deformed cylinder, b-box, p-prism with a triangle base.**

| Site | Segment | Shape | h | a | b | c | d | Ct.Th | Face, covered by Ct.Th | BV/TV | Tb.Th | σTb.Th | Tb.Sp | σTb.Sp |
|---|---|---|---|---|---|---|---|---|---|---|---|---|---|---|
| Clavicle | Acromial end | dc | 17 | 19 | 11 | 11 | 8.3 | 0.8 | lateral | 0.29 | 0.15 | 22 | 0.8 | 23 |
| Clavicle | Body | c | 30 | 11 | 8.3 | | | 1.8 | lateral | 0.15 | 0.197 | 22 | 0.8 | 23 |
| Clavicle | Sternal end | dc | 17 | 22 | 20 | 11 | 8.3 | 0.8 | lateral | 0.29 | 0.15 | 22 | 0.8 | 23 |
| Femur | Distal end | dc | 69 | 78 | 33 | 21 | 21 | 1.1 | lateral | 0.26 | 0.24 | 22 | 0.538 | 23 |
| Femur | Lower proximal end | c | 30 | 25 | 25 | | | 2.2 | lateral | 0.35 | 0.24 | 22 | 0.538 | 23 |
| Femur | Upper proximal end | c | 30 | 25 | 25 | | | 1.8 | lateral | 0.35 | 0.24 | 22 | 0.538 | 23 |
| Humery | Distal end | dc | 27 | 54 | 18 | 18 | 18 | 0.8 | lateral | 0.22 | 0.208 | 22 | 0.58 | 23 |
| Humery | Proximal end | dc | 27 | 39 | 33 | 18 | 18 | 1.1 | lateral | 0.22 | 0.208 | 22 | 0.58 | 23 |
| Pelvis | Ilium acetabular part | dc | 26 | 44 | 20 | 37 | 28 | 0.9 | lateral | 0.25 | 0.155 | 10 | 0.459 | 10 |
| Pelvis | Ilium flat part 1 | b | 8 | 30 | 30 | | | 1.7[1] ab1<br>0.9[1] ab2 | | 0.25 | 0.155 | 10 | 0.459 | 10 |
| Pelvis | Ilium flat part 2 | b | 8 | 30 | 30 | | | 0.9 | ab | 0.25 | 0.155 | 10 | 0.459 | 10 |
| Pelvis | Ischial ramus inferior | c | 32 | 11 | 11 | | | 0.5 | lateral | 0.25 | 0.155 | 10 | 0.6 | 10 |
| Pelvis | Ischium acetabular part | p | 29 | 29 | 28 | | | 0.5 | ah; bh; ch | 0.25 | 0.155 | 10 | 0.6 | 10 |
| Pelvis | Ischium tuberosity | b | 34 | 19 | 19 | | | 0.5 | ha; hb | 0.25 | 0.155 | 10 | 0.6 | 10 |
| Pelvis | Pubis acetabular part | dc | 9.8 | 28 | 21 | 16 | 11 | 0.5 | lateral | 0.25 | 0.155 | 10 | 0.6 | 10 |
| Pelvis | Pubis ramus inferior | c | 32 | 11 | 11 | | | 0.5 | lateral | 0.25 | 0.155 | 10 | 0.6 | 10 |
| Pelvis | Pubis ramus superior | c | 39 | 16 | 11 | | | 0.5 | lateral | 0.25 | 0.155 | 10 | 0.6 | 10 |
| Ribs | Ribs | b | 10 | 30 | 5 | | | 0.7 | ha; ab | 0.2 | 0.231 | 22 | 0.505 | 23 |
| Sacrum | Body-ala 1 | b | 22 | 88 | 26 | | | 0.9 | ha | 0.14 | 0.118 | 48 | 0.65 | 43 |
| Sacrum | Body-ala 2 | b | 20 | 70 | 15 | | | 0.9 | ha | 0.14 | 0.118 | 48 | 0.65 | 43 |
| Sacrum | Body-ala 3 | b | 18 | 62 | 10 | | | 0.9 | ha | 0.14 | 0.118 | 48 | 0.65 | 43 |
| Sacrum | Body-ala 4 | b | 13 | 53 | 7.7 | | | 0.9 | ha | 0.14 | 0.118 | 48 | 0.65 | 43 |
| Sacrum | Body-ala 5 | b | 13 | 44 | 7.7 | | | 0.9 | ha | 0.14 | 0.118 | 48 | 0.65 | 43 |
| Scapula | Acromion | b | 8.2 | 27 | 21 | | | 0.8 | ha; hb1; ab | 0.22 | 0.24 | 22 | 0.964 | 23 |
| Scapula | Glenoid | c | 15 | 28 | 20 | | | 0.9 | lateral | 0.22 | 0.24 | 22 | 0.964 | 23 |
| Scapula | Lateral margin | b | 30 | 3.5 | 10 | | | 0.8 | ha1; hb | 0.22 | 0.24 | 22 | 0.964 | 23 |
| Skull | Flat bones | b | 4.6 | 30 | 30 | | | 1.2 | ab | 0.52 | 0.29 | 8 | 0.57 | 15 |
| Sternum | Sternum | b | 8.5 | 30 | 30 | | | 0.7 | ab | 0.15 | 0.15 | 22 | 1.0 | 23 |
| Tibia and fibula | Distal end | dc | 48 | 35 | 35 | 21 | 21 | 0.7 | lateral | 0.25 | 0.212 | 22 | 0.735 | 23 |
| Tibia and fibula | Fibula ends | c | 30 | 11 | 11 | | | 1.7 | lateral | 0.25 | 0.212 | 22 | 0.735 | 23 |
| Tibia and fibula | Proximal end | dc | 49 | 63 | 32 | 21 | 21 | 0.7 | lateral | 0.25 | 0.212 | 22 | 0.735 | 23 |
| Vertebra | C-body | c | 9.4 | 19 | 14 | | | 0.2 | lateral | 0.21 | 0.14 | 48 | 0.65 | 43 |
| Vertebra | L-body | c | 19 | 36 | 27 | | | 0.2 | lateral | 0.14 | 0.118 | 48 | 0.65 | 43 |
| Vertebra | L-spinous proc | b | 17 | 27 | 5.2 | | | 0.2 | ha; ab; hb1 | 0.14 | 0.118 | 48 | 0.65 | 43 |
| Vertebra | L-transverse proc | b | 8.6 | 16 | 5.2 | | | 0.2 | ha; ab; hb1 | 0.14 | 0.118 | 48 | 0.65 | 43 |
| Vertebra | T-body | c | 14 | 27 | 22 | | | 0.2 | lateral | 0.14 | 0.118 | 48 | 0.65 | 43 |
| Vertebra | T-spinous proc | b | 7.2 | 25 | 4.1 | | | 0.2 | ha; ab; hb1 | 0.14 | 0.118 | 48 | 0.65 | 43 |
| Vertebra | T-transverse proc | b | 8.6 | 13 | 7.3 | | | 0.2 | ha; ab; hb1 | 0.14 | 0.118 | 48 | 0.65 | 43 |

**Table 11. Parameters of bone segment phantoms for a reference 15-year-old male; macro-(*h, a, b, c, d, Ct.Th*) and microarchitecture (*Tb.Th, Tb.Sp*) values are given in mm; *BV/TV* is in relative units; variability ($\sigma_{Tb.Th}$ and $\sigma_{Tb.Sp}$) is given in percentages; the shape of a segment is designated as follows: c-cylinder, dc-deformed cylinder, b-box, p-prism with a triangle base, t-tube.**

| Site | Segment | Shape | h | a | b | c | d | Ct. Th | Face, covered by Ct.Th | BV/TV | Tb.Th | $\sigma Tb. Th$ | Tb.Sp | $\sigma Tb. Sp$ |
|---|---|---|---|---|---|---|---|---|---|---|---|---|---|---|
| Clavicle | Acromial end | dc | 19.8 | 22 | 12 | 12 | 12 | 0.8 | lateral | 0.29 | 0.15 | 22 | 0.8 | 23 |
| Clavicle | Shaft | c | 30 | 12 | 12 | | | 1.8 | lateral | 0.15 | 0.20 | 22 | 0.8 | 23 |
| Clavicle | Sternal end | dc | 19.8 | 26 | 24 | 12 | 12 | 0.8 | lateral | 0.29 | 0.15 | 22 | 0.8 | 23 |
| Femur | Neck | c | 29.6 | 36 | 32 | | | 2 | lateral | 0.35 | 0.24 | 22 | 0.54 | 23 |
| Femur | Trochanter area | dc | 41 | 66 | 44 | 30 | 30 | 2.3 | lateral | 0.26 | 0.24 | 22 | 0.54 | 23 |
| Humerus | Proximal end | dc | 28 | 56 | 56 | 19 | 19.4 | 1.1 | lateral | 0.22 | 0.21 | 22 | 0.58 | 23 |
| Pelvis | Acetabulum | t | 29 | 26 | 21 | | | 0.5 3.6 | lateral ab1 | 0.25 | 0.16 | 10 | 0.46 | 10 |
| Pelvis | Iliac ala | b | 9.5 | 30 | 30 | | | 1 | ab | 0.25 | 0.16 | 10 | 0.46 | 10 |
| Pelvis | Iliac crest | b | 11 | 30 | 13 | | | 1 | ha; ab1 | 0.25 | 0.16 | 10 | 0.46 | 10 |
| Pelvis | Iliac dorsal seg. | b | 19 | 30 | 30 | | | 1 | ab | 0.25 | 0.16 | 10 | 0.46 | 10 |
| Pelvis | Ischium ramus | c | 30 | 34 | 25 | | | 0.5 | lateral | 0.25 | 0.16 | 10 | 0.75 | 10 |
| Pelvis | Pubis ramus inf. | dc | 47 | 16 | 22 | 26 | 14 | 0.5 | lateral | 0.25 | 0.16 | 10 | 0.75 | 10 |
| Pelvis | Pubis ramus superior (low) | b | 32 | 15 | 29 | | | 0.7 1.5 | hb1; ha1 ab1 | 0.25 | 0.16 | 10 | 0.75 | 10 |
| Pelvis | Pubis ramus superior (upper) | b | 51.2 | 14.5 | 16 | | | 0.7 1.5 | ha; hb1 ab1 | 0.25 | 0.16 | 10 | 0.75 | 10 |
| Ribs | 1, 2 | b | 17 | 30 | 7 | | | 0.7 | ha; ab | 0.12 | 0.15 | 22 | 0.82 | 23 |
| Ribs | 11, 12 | b | 11 | 30 | 6 | | | 0.7 | ha; ab | 0.12 | 0.15 | 22 | 0.82 | 23 |
| Ribs | 3, 4, 9, 10 | b | 13 | 30 | 7 | | | 1.2 | ha; ab | 0.12 | 0.15 | 22 | 0.82 | 23 |
| Ribs | 5,6,7,8 | b | 14 | 30 | 8 | | | 0.7 | ha; ab | 0.12 | 0.15 | 22 | 0.82 | 23 |
| Sacrum | Ala 3–4 | p | 19 | 18 | 38.5 | | | 1.2 | bh; ch; abb1 | 0.14 | 0.12 | 48 | 0.65 | 43 |
| Sacrum | Body 1 | b | 30 | 40 | 24.5 | | | 1.2 | ha | 0.14 | 0.12 | 48 | 0.65 | 43 |
| Sacrum | Body 2–3 | b | 46 | 28.7 | 15 | | | 1.2 | ha | 0.14 | 0.12 | 48 | 0.65 | 43 |
| Sacrum | Body 4–5 | b | 36 | 28 | 8 | | | 1.2 | ha | 0.14 | 0.12 | 48 | 0.65 | 43 |
| Sacrum | Pedicle 1 | c | 13.9 | 23.7 | 15.3 | | | 1.2 | lateral | 0.14 | 0.12 | 48 | 0.65 | 43 |
| Sacrum | Pedicle 2 | c | 14.2 | 25 | 13.6 | | | 1.2 | lateral | 0.14 | 0.12 | 48 | 0.65 | 43 |
| Sacrum | Pedicle 3 | c | 13.9 | 18.3 | 13.2 | | | 1.2 | lateral | 0.14 | 0.12 | 48 | 0.65 | 43 |
| Sacrum | Pedicle 4 | c | 13.9 | 14.5 | 11.2 | | | 1.2 | lateral | 0.14 | 0.12 | 48 | 0.65 | 43 |
| Sacrum | Sacral ala 1 | b | 30 | 20 | 42 | | | 1.2 | ha; ab1 | 0.14 | 0.12 | 48 | 0.65 | 43 |
| Sacrum | Sacral ala 2 | b | 26 | 23 | 25 | | | 1.2 | ha | 0.14 | 0.12 | 48 | 0.65 | 43 |
| Scapula | Acromion | b | 8.8 | 32.4 | 25.2 | | | 0.8 | ha; hb1; ab | 0.22 | 0.24 | 22 | 0.96 | 23 |
| Scapula | Glenoid | c | 16.9 | 30.7 | 22.3 | | | 0.9 | lateral | 0.22 | 0.24 | 22 | 0.96 | 23 |
| Scapula | Lateral margin | b | 30 | 3.5 | 10 | | | 0.8 | ha1; hb | 0.22 | 0.24 | 22 | 0.96 | 23 |
| Skull | Flat bones | b | 5.2 | 30 | 30 | | | 1.3 | ab | 0.52 | 0.29 | 8 | 0.57 | 15 |
| Sternum | Sternum | b | 10.6 | 30 | 30 | | | 0.9 | ab | 0.15 | 0.15 | 22 | 1 | 23 |
| Vertebra | C1 mass | b | 15 | 19 | 13.3 | | | 1.3 | ha; ab1 | 0.21 | 0.15 | 48 | 0.5 | 43 |
| Vertebra | C2-body | b | 19.2 | 14.3 | 17.5 | | | 0.3 | hb | 0.21 | 0.15 | 48 | 0.5 | 43 |
| Vertebra | C3-7 body | c | 11.9 | 15.2 | 19 | | | 0.3 | lateral | 0.21 | 0.14 | 48 | 0.6 | 43 |
| Vertebra | L- body | c | 24.7 | 44.5 | 31.5 | | | 0.4 | lateral | 0.14 | 0.12 | 48 | 0.65 | 43 |
| Vertebra | L- lamina+inf.pr | b | 20.4 | 12.7 | 4.1 | | | 1 | ha; ab1 | 0.15 | 0.1 | 48 | 0.6 | 43 |
| Vertebra | L- transverse p. | b | 10.7 | 20.5 | 5.6 | | | 0.4 | ha; ab; hb1 | 0.14 | 0.12 | 48 | 0.65 | 43 |
| Vertebra | L-spinous p. | b | 24 | 31 | 5.6 | | | 0.4 | ha; ab; hb1 | 0.14 | 0.12 | 48 | 0.65 | 43 |
| Vertebra | L-superior p. | b | 14 | 15 | 12 | | | 1 | ha; ab; hb1 | 0.15 | 0.1 | 48 | 0.6 | 43 |

*(Continued)*

**Table 11.** (Continued)

| Site | Segment | Shape | h | a | b | c | d | Ct. Th | Face, covered by Ct.Th | BV/TV | Tb.Th | σTb. Th | Tb.Sp | σTb. Sp |
|------|---------|-------|---|---|---|---|---|--------|------------------------|-------|-------|---------|-------|---------|
| Vertebra | T- body | c | 18.6 | 28.7 | 24.9 | | | 0.4 | lateral | 0.14 | 0.12 | 48 | 0.65 | 43 |
| Vertebra | T- lamina+inf. | b | 32 | 10.2 | 4.2 | | | 1.3 | ha; ab1 | 0.16 | 0.15 | 48 | 0.6 | 43 |
| Vertebra | T- transverse p. | b | 9.9 | 14.8 | 8.6 | | | 0.4 | ha; ab; hb1 | 0.14 | 0.12 | 48 | 0.65 | 43 |
| Vertebra | T-spinous p. | b | 9.3 | 33 | 4.9 | | | 0.4 | ha; ab; hb1 | 0.14 | 0.12 | 48 | 0.65 | 43 |
| Vertebra | T-superior pr. | b | 11.4 | 11.3 | 4.4 | | | 1.3 | ha; ab; hb1 | 0.16 | 0.15 | 48 | 0.6 | 43 |

The bases of simple and deformed cylinders are not covered by cortical layers, and *Ct.Th* values refer only to the lateral surface. Boxes and triangular prisms may also have only part of the surfaces covered. Therefore, to describe the phantoms, it was necessary to introduce a designation for the faces covered with cortical bone. Rectangular faces of boxes are designated as *ha*, *hb* and *ab* in accordance with the edges $h \times a$, $h \times b$ and $a \times b$. Triangular bases of a prism are designated as *abb.* All these designations refer to two parallel faces. If the parallel faces are covered with a cortical layer of different thicknesses, then the number 1 or 2 is added to the face index (for example, *ab1* and *ab2*). Because the faces are mirror-symmetrical, it is not important for dosimetric modelling to specify which of the sides is covered with the cortical layer.

Tables 7–14 present the parameters of reference phantoms. Characteristics of individual variability of the parameters are presented in Supplementary Material (S1 File). Characteristics of segment-specific volumes of trabecular bone, cortical bone, and bone marrow are presented in Supplementary Material (S2 File). Individual variability of morphometric parameters and its impact on dose formation was discussed in detail in [21,22]. The voxel resolution of the segment-specific phantoms (see S1 File) ranges from 50 μm to 200 μm depending on *Tb.Th*. Half of the phantoms were generated with a voxel resolution of 70 μm to 110 μm.

Because the bone sizes change with age, some of the segments become too large to be modeled entirely with the required voxel resolution. Therefore, they are truncated. For example, the truncated segment representing the adult ilium (shown in blue in Fig 1) is described by a box with a surface area of $9\,cm^2$, while the surface area of the entire segment reaches $91.2\,cm^2$. Similar age-related changes are typical for other parts of the skeleton. Media volumes (cortical bone, trabecular bone, and bone marrow) of a truncated model were recalculated to volumes of whole segment using a factor ($k_s$) which is proportionate to either linear dimensions or surface area (depending on the method of truncating). In addition, many segments are paired and therefore total volumes of segment-specific media were corrected to the number of similar models ($n_s$). Corrected segment-specific media volumes are presented in Supplementary Material (S2 File). The range of individual variability of medium volumes was calculated based on the random generation of a set of segment-specific phantoms. To simulate individual variability, computer code "Trabecula" assumes a positive correlation of normally distributed linear dimensions of bone segments and a negative correlation of lognormally distributed *Tb.Sp* and *Tb.Th*; *Ct.Th* is assumed to be a normally distributed independent parameter [20].

## Discussion

Our work to define bone geometry and bone segment parameters resulted in the development of phantoms to improve the bone marrow dosimetry. AM is a critical organ of hematopoiesis, therefore AM dosimetry is extremely important for clarifying the risks of hematological radiation effects. The representative computational phantoms are needed to produce accurate estimates of AM internal doses.

Some simplifications were introduced to the modeling of bones of complex shape. The validity of bone shape and segmentation approaches were carefully checked. For instance, according to [52], the volume fraction of a lumbar vertebral body makes up $0.65 \pm 0.04$ (mean±SD) of the lumbar vertebra of adults. The volume fraction of a vertebral body in the SPSD phantom is $0.68 \pm 0.05$. These values are in good agreement with each other.

**Table 12. Parameters of bone segment phantoms for a reference 15-year-old female; macro-(*h, a, b, c, d, Ct.Th*) and microarchitecture (*Tb.Th*, *Tb.Sp*) values are given in mm; *BV/TV* is in relative units; variability ($\sigma_{Tb.Th}$ and $\sigma_{Tb.Sp}$) is given in percentages; the shape of a segment is designated as follows: c-cylinder, dc-deformed cylinder, b-box, p-prism with a triangle base, t-tube.**

| Site | Segment | Shape | h | a | b | c | d | Ct.Th | Face, covered by Ct.Th | BV/TV | Tb.Th | σTb.Th | Tb.Sp | σTb.Sp |
|---|---|---|---|---|---|---|---|---|---|---|---|---|---|---|
| Clavicle | Acromial end | dc | 20.7 | 21 | 10 | 10 | 10 | 0.8 | lateral | 0.29 | 0.15 | 22 | 0.8 | 23 |
| Clavicle | Shaft | c | 30 | 10 | 10 | | | 1.8 | lateral | 0.15 | 0.20 | 22 | 0.8 | 23 |
| Clavicle | Sternal end | dc | 20.7 | 24 | 21 | 10 | 10 | 0.8 | lateral | 0.29 | 0.15 | 22 | 0.8 | 23 |
| Femur | Neck | c | 30.5 | 29.4 | 23.9 | | | 2 | lateral | 0.35 | 0.24 | 22 | 0.54 | 23 |
| Femur | Trochanter area | dc | 34 | 58 | 39 | 27 | 27 | 2.3 | lateral | 0.26 | 0.24 | 22 | 0.54 | 23 |
| Humerus | Proximal end | dc | 24.9 | 51.3 | 51.3 | 19.4 | 19.4 | 1.1 | lateral | 0.22 | 0.21 | 22 | 0.58 | 23 |
| Pelvis | Acetabulum | t | 29 | 26 | 21 | | | 0.5 3.6 | lateral [3] ab1 | 0.25 | 0.16 | 10 | 0.46 | 10 |
| Pelvis | Iliac ala | b | 9.5 | 30 | 30 | | | 1 | ab | 0.25 | 0.16 | 10 | 0.46 | 10 |
| Pelvis | Iliac crest | b | 11 | 30 | 13 | | | 1 | ha; ab1 | 0.25 | 0.16 | 10 | 0.46 | 10 |
| Pelvis | Iliac dorsal seg. | b | 19 | 30 | 30 | | | 1 | ab | 0.25 | 0.16 | 10 | 0.46 | 10 |
| Pelvis | Ischium ramus | c | 30 | 34 | 25 | | | 0.5 | lateral | 0.25 | 0.16 | 10 | 0.75 | 10 |
| Pelvis | Pubis ramus inf. | dc | 47 | 16 | 22 | 26 | 14 | 0.5 | lateral | 0.25 | 0.16 | 10 | 0.75 | 10 |
| Pelvis | Pubis ramus superior (Low) | b | 31 | 14 | 33 | | | 0.7 1.5 | hb; ha1 ab1 | 0.25 | 0.16 | 10 | 0.75 | 10 |
| Pelvis | Pubis ramus superior (Upper) | b | 83 | 11 | 16 | | | 0.7 1.5 | ha; hb1 ab1 | 0.25 | 0.16 | 10 | 0.75 | 10 |
| Ribs | 1, 2 | b | 17 | 30 | 7 | | | 0.7 | ha; ab | 0.12 | 0.15 | 22 | 0.82 | 23 |
| Ribs | 11, 12 | b | 11 | 30 | 6 | | | 0.7 | ha; ab | 0.12 | 0.15 | 22 | 0.82 | 23 |
| Ribs | 3, 4, 9, 10 | b | 13 | 30 | 7 | | | 0.7 | ha; ab | 0.12 | 0.15 | 22 | 0.82 | 23 |
| Ribs | 5,6,7,8 | b | 14 | 30 | 8 | | | 0.7 | ha; ab | 0.12 | 0.15 | 22 | 0.82 | 23 |
| Sacrum | Ala 3–4 | p | 19 | 18 | 38.5 | | | 1.2 | bh; ch; abb1 | 0.14 | 0.12 | 48 | 0.65 | 43 |
| Sacrum | Body 1 | b | 30 | 37.8 | 22.2 | | | 1.2 | ha | 0.14 | 0.12 | 48 | 0.65 | 43 |
| Sacrum | Body 2–3 | b | 45.2 | 28 | 13.8 | | | 1.2 | ha | 0.14 | 0.12 | 48 | 0.65 | 43 |
| Sacrum | Body 4–5 | b | 35 | 28 | 8.5 | | | 1.2 | ha | 0.14 | 0.12 | 48 | 0.65 | 43 |
| Sacrum | Pedicle 1 | c | 13.9 | 23.7 | 15.3 | | | 1.2 | lateral | 0.14 | 0.12 | 48 | 0.65 | 43 |
| Sacrum | Pedicle 2 | c | 14.2 | 25 | 13.6 | | | 1.2 | lateral | 0.14 | 0.12 | 48 | 0.65 | 43 |
| Sacrum | Pedicle 3 | c | 13.9 | 18.3 | 13.2 | | | 1.2 | lateral | 0.14 | 0.12 | 48 | 0.65 | 43 |
| Sacrum | Pedicle 4 | c | 13.9 | 14.5 | 11.2 | | | 1.2 | lateral | 0.14 | 0.12 | 48 | 0.65 | 43 |
| Sacrum | Sacral ala 1 | b | 30 | 21 | 38.6 | | | 1.2 | ha; ab1 | 0.14 | 0.12 | 48 | 0.65 | 43 |
| Sacrum | Sacral ala 2 | b | 26 | 23 | 22.7 | | | 1.2 | ha | 0.14 | 0.12 | 48 | 0.65 | 43 |
| Scapula | Acromion | b | 8.8 | 32.4 | 25.2 | | | 0.8 | ha; hb1; ab | 0.22 | 0.24 | 22 | 0.96 | 23 |
| Scapula | Glenoid | c | 16.9 | 30.7 | 22.3 | | | 0.9 | lateral | 0.22 | 0.24 | 22 | 0.96 | 23 |
| Scapula | Lateral margin | b | 30 | 3.5 | 10 | | | 0.8 | ha1; hb | 0.22 | 0.24 | 22 | 0.96 | 23 |
| Skull | Flat bones | b | 5.2 | 30 | 30 | | | 1.3 | ab | 0.52 | 0.29 | 8 | 0.57 | 15 |
| Sternum | Sternum | b | 10.6 | 30 | 30 | | | 0.9 | ab | 0.15 | 0.15 | 22 | 1 | 23 |
| Vertebra | C1 mass | b | 15 | 19 | 13.3 | | | 1.3 | ha; ab1 | 0.21 | 0.15 | 48 | 0.5 | 43 |
| Vertebra | C2-body | b | 19.2 | 14.3 | 17.5 | | | 0.3 | hb | 0.21 | 0.15 | 48 | 0.5 | 43 |
| Vertebra | C3-7 body | c | 11.9 | 15.2 | 19 | | | 0.3 | lateral | 0.21 | 0.14 | 48 | 0.6 | 43 |
| Vertebra | L- body | c | 24.7 | 44.5 | 31.5 | | | 0.4 | lateral | 0.14 | 0.12 | 48 | 0.65 | 43 |
| Vertebra | L- lamina+inf.pr | b | 20.4 | 12.7 | 4.1 | | | 1 | ha; ab1 | 0.15 | 0.1 | 48 | 0.6 | 43 |
| Vertebra | L- transverse p. | b | 10.7 | 20.5 | 5.6 | | | 0.4 | ha; ab; hb1 | 0.14 | 0.12 | 48 | 0.65 | 43 |
| Vertebra | L-spinous p. | b | 24 | 31 | 5.6 | | | 0.4 | ha; ab; hb1 | 0.14 | 0.12 | 48 | 0.65 | 43 |
| Vertebra | L-superior p. | b | 14 | 15 | 12 | | | 1 | ha; ab; hb1 | 0.15 | 0.1 | 48 | 0.6 | 43 |

*(Continued)*

**Table 12.** (Continued)

| Site | Segment | Shape | h | a | b | c | d | Ct.Th | Face, covered by Ct.Th | BV/TV | Tb.Th | σTb.Th | Tb.Sp | σTb.Sp |
|------|---------|-------|---|---|---|---|---|-------|------------------------|-------|-------|--------|-------|--------|
| Vertebra | T- body | c | 18.6 | 28.7 | 24.9 | | | 0.4 | lateral | 0.14 | 0.12 | 48 | 0.65 | 43 |
| Vertebra | T- lamina+inf. | b | 32 | 10.2 | 4.2 | | | 1.3 | ha; ab1 | 0.16 | 0.15 | 48 | 0.6 | 43 |
| Vertebra | T- transverse p. | b | 9.9 | 14.8 | 8.6 | | | 0.4 | ha; ab; hb1 | 0.14 | 0.12 | 48 | 0.65 | 43 |
| Vertebra | T-spinous p. | b | 9.3 | 33 | 4.9 | | | 0.4 | ha; ab; hb1 | 0.14 | 0.12 | 48 | 0.65 | 43 |
| Vertebra | T-superior pr. | b | 11.4 | 11.3 | 4.4 | | | 1.3 | ha; ab; hb1 | 0.16 | 0.15 | 48 | 0.6 | 43 |

The SPSD method considers the hematopoietic skeleton segments only. Not all the sites of the skeleton under consideration are hematopoietic throughout human life (Table 2). However, all ossified bones of the newborn skeleton were modeled because they are all hematopoietic. To test the adequacy of the SPSD modeling approach, we compared the masses of SPSD phantoms with measurements of real bones. For example, a comparison of average bone masses of 40 full-term newborns and SPSD masses of corresponding bone phantoms (calculated as the sum of the masses of the bone segments) is shown in Fig 5 [53]. As shown in Fig 5, the masses of SPSD phantoms are in good agreement with measurement results.

The volume and hematopoietic status may change with age for individual segments within a given bone site. Fig 6 demonstrates the age dependence of the total volumes of phantoms of thoracic and lumbar vertebrae body, the proximal humerus and scapula glenoid.

The modeling results presented in Fig 6 are in good agreement with the age-related characteristics of the anatomy and physiology of the human skeletal system. Fig 7A presents a comparison of bone masses of the SPSD phantom (clavicle, sternum, ribs, sacrum, as well as cervical, thoracic and lumbar vertebrae) of adult man with corresponding wet masses of ICRP reference man [17]. We have also included available data on the bone masses of the image-based phantom of VIP-man into Fig 7A [54].

As shown in Fig 7A, literature-derived bone masses of ICRP reference man are mainly consistent with literature-based SPSD phantom (at least within the individual variability). A significant difference is found for cervical vertebrae. The masses of SPSD phantoms of the cervical vertebrae are 4.5 times less than the masses of both ICRP reference man and VIP-man. Comparisons of only spongiosa masses in SPSD and ICRP phantoms (both voxel and mesh-type) of adult man [4,6] demonstrate a similar tendency of good agreement, except for the cervical vertebrae (Fig 7B).

The difference in spongiosa mass can be due to a difference in *BV/TV* (which is proportional to spongiosa density) or/and bone sizes. Table 15 presents a comparison of *BV/TV* in the bone sites described using SPSD and ICRP 70. Note that the SPSD approach is based on a vast amount of new data including μCT or MRI-data of spongiosa (as compared to ICRP 70 data) [31].

The 90% CI of SPSD estimates was used as a comparison interval. Grey rows in Table 15 correspond to the bone sites for which the value provided by ICRP 70 does not fall within the 90% CI of the SPSD model. As a result, 5 of 12 bone sites have some discrepancy in *BV/TV* between the two estimates, including cervical vertebrae and clavicles. The clavicle *BV/TV* is about two times greater in SPSD than in ICRP. This is the most probable reason for a greater spongiosa mass in SPSD (Fig 7B). However, the SPSD cervical vertebrae *BV/TV* is also about two times greater, and the reason for the lower ICRP mass could be the difference in size of the models and number of the modelled parts of the cervical vertebrae in the SPSD. However, the whole-skeleton AM fraction in the adult cervical vertebrae is low (−3.5%), which means that this site ultimately has little effect on the skeletal-average dose to AM.

We have also compared *Ct.Th* for the newborn and 15-year-old in SPSD phantoms and University of Florida image-based phantoms (UF phantoms) [2]. *Ct.Th* values in the SPSD phantoms were on average 1.9 times greater. This may be due to the difference in the methods of parameter estimation. The advantage of the SPSD segmentation-based method is

**Table 13. Parameters of bone segment phantoms for a reference adult male; macro-architecture (*h, a, b, c, d, Ct.Th*) and microarchitecture (*Tb.Th, Tb.Sp*) values are given in mm; *BV/TV* is in relative units; variability ($\sigma_{Tb.Th}$ and $\sigma_{Tb.Sp}$) is given in percentages; the shape of a segment is designated as follows: c-cylinder, dc-deformed cylinder, b-box, p-prism with a triangle base, t-tube.**

| Site | Segment | Shape | h | a | b | c | d | Ct.Th | Face, covered by Ct.Th | BV/TV | Tb.Th | σTb.Th | Tb.Sp | σTb.Sp |
|---|---|---|---|---|---|---|---|---|---|---|---|---|---|---|
| Clavicle | Acromial shaft | c | 56 | 26 | 24 | 12 | 12 | 1.8 | lateral | 0.13 | 0.19 | 22 | 0.8 | 23 |
| Clavicle | Ends | dc | 20 | 26 | 240 | | | 0.6 | lateral | 0.29 | 0.14 | 22 | 0.8 | 23 |
| Clavicle | Sternal shaft | dc | 56 | 22 | 12 | 12 | 12 | 1.8 | lateral | 0.13 | 0.19 | 22 | 0.8 | 23 |
| Femur | Neck | c | 30 | 36 | 32 | | | 1.9 | lateral | 0.17 | 0.19 | 22 | 0.78 | 23 |
| Femur | Trochanter area | dc | 43 | 66 | 44 | 30 | 30 | 2.3 | lateral | 0.11 | 0.136 | 22 | 0.99 | 23 |
| Humerus | Proximal end | dc | 28 | 56 | 56 | 25 | 25 | 1.1 | lateral | 0.06 | 0.1 | 22 | 2.37 | 23 |
| Pelvis | Acetabulum | t | 29 | 26 | 21 | | | 0.5 3.6 | lateral ab1 | 0.19 | 0.13 | 10 | 0.6 | 10 |
| Pelvis | Iliac ala | b | 9.5 | 30 | 30 | | | 1 | ab | 0.19 | 0.13 | 10 | 0.6 | 10 |
| Pelvis | Iliac crest | b | 11 | 30 | 13 | | | 1 | ha; ab1 | 0.19 | 0.13 | 10 | 0.6 | 10 |
| Pelvis | Iliac dorsal segment | b | 19 | 30 | 30 | | | 1 | ab | 0.19 | 0.13 | 10 | 0.6 | 10 |
| Pelvis | Ischium ramus | c | 30 | 34 | 25 | | | 0.5 | lateral | 0.25 | 0.3 | 10 | 1 | 10 |
| Pelvis | Pubis ramus inferior | dc | 47 | 16 | 22 | 26 | 14 | 0.5 | lateral | 0.25 | 0.3 | 10 | 1 | 10 |
| Pelvis | Pubis ramus superior (lower) | b | 32 | 15 | 29 | | | 0.7 1.5 | hb; ha1 ab1 | 0.17 | 0.29 | 10 | 1 | 10 |
| Pelvis | Pubis ramus superior (upper) | b | 51.2 | 14.5 | 16 | | | 0.7 1.5 | ha; hb1 ab1 | 0.17 | 0.29 | 10 | 1 | 10 |
| Ribs | 1, 2 | b | 17 | 30 | 7 | | | 0.7 | ha; ab | 0.12 | 0.147 | 22 | 0.82 | 23 |
| Ribs | 11, 12 | b | 11 | 30 | 6 | | | 0.7 | ha; ab | 0.12 | 0.147 | 22 | 0.82 | 23 |
| Ribs | 3, 4, 9, 10 | b | 13 | 30 | 7 | | | 0.7 | ha; ab | 0.12 | 0.147 | 22 | 0.82 | 23 |
| Ribs | 5,6,7,8 | b | 14 | 30 | 8 | | | 0.7 | ha; ab | 0.12 | 0.147 | 22 | 0.82 | 23 |
| Sacrum | Ala 3–4 | p | 19 | 18 | 38.5 | | | 1.5 | bh; ch; abb1 | 0.15 | 0.1 | 48 | 0.6 | 43 |
| Sacrum | Body 1 | b | 30 | 40 | 24.5 | | | 1.5 | ha | 0.15 | 0.1 | 48 | 0.6 | 43 |
| Sacrum | Body 2–3 | b | 46 | 28.7 | 15 | | | 1.5 | ha | 0.15 | 0.1 | 48 | 0.6 | 43 |
| Sacrum | Body 4–5 | b | 36 | 28 | 8.5 | | | 1.5 | ha | 0.15 | 0.1 | 48 | 0.6 | 43 |
| Sacrum | Pedicle 1 | c | 13.9 | 23.7 | 15.3 | | | 1.5 | lateral | 0.15 | 0.1 | 48 | 0.6 | 43 |
| Sacrum | Pedicle 2 | c | 14.2 | 25 | 13.6 | | | 1.5 | lateral | 0.15 | 0.1 | 48 | 0.6 | 43 |
| Sacrum | Pedicle 3 | c | 13.9 | 18.3 | 13.2 | | | 1.5 | lateral | 0.15 | 0.1 | 48 | 0.6 | 43 |
| Sacrum | Pedicle 4 | c | 13.9 | 14.5 | 11.2 | | | 1.5 | lateral | 0.15 | 0.1 | 48 | 0.6 | 43 |
| Sacrum | Sacral ala 1 | b | 30 | 20 | 42 | | | 1.5 | ha; ab1 | 0.15 | 0.1 | 48 | 0.6 | 43 |
| Sacrum | Sacral ala 2 | b | 26 | 23 | 25 | | | 1.5 | ha | 0.15 | 0.1 | 48 | 0.6 | 43 |
| Scapula | Acromion | b | 8.8 | 48 | 26 | | | 0.8 | ha, hb1; ab | 0.22 | 0.24 | 22 | 0.96 | 23 |
| Scapula | Glenoid | c | 20 | 36 | 26 | | | 0.9 | lateral | 0.22 | 0.24 | 22 | 0.96 | 23 |
| Scapula | Lateral margin | b | 30 | 5 | 10 | | | 0.8 | ha1; hb | 0.22 | 0.24 | 22 | 0.96 | 23 |
| Skull | Flat bones | b | 5.2 | 30 | 30 | | | 1.3 1.5 | ab1 ab2 | 0.52 | 0.29 | 8 | 0.57 | 15 |
| Sternum | Manubrium | b | 1.3 | 30 | 30 | | | 1.45 | ab | 0.15 | 0.15 | 22 | 1.4 | 23 |
| Sternum | Sternum bdoy | b | 1.0 | 30 | 30 | | | 1.1 | ab | 0.15 | 0.15 | 22 | 1.4 | 23 |
| Vertebra | C-body 3–7 | c | 13 | 19 | 16 | | | 0.3 | lateral | 0.21 | 0.15 | 48 | 0.5 | 43 |
| Vertebra | Cervical body 2 | b | 19.2 | 14.3 | 17.5 | | | 0.3 | hb | 0.21 | 0.15 | 48 | 0.5 | 43 |
| Vertebra | Cervical lateral 1 | b | 15 | 11.4 | 10.5 | | | 0.3 | ha; ab1 | 0.21 | 0.15 | 48 | 0.5 | 43 |
| Vertebra | L- lamina+inf.pr | b | 20.4 | 12.7 | 4.1 | | | 1 | ha; ab1 | 0.15 | 0.1 | 48 | 0.6 | 43 |
| Vertebra | L-body | c | 27 | 35 | 47 | | | 1.3 | lateral | 0.16 | 0.15 | 48 | 0.6 | 43 |
| Vertebra | L-spinous pr. | b | 24 | 31 | 6 | | | 0.4 | ha; ab; hb1 | 0.15 | 0.1 | 48 | 0.6 | 43 |

*(Continued)*

**Table 13.** (Continued)

| Site | Segment | Shape | h | a | b | c | d | Ct.Th | Face, covered by Ct.Th | BV/TV | Tb.Th | σTb.Th | Tb.Sp | σTb.Sp |
|------|---------|-------|---|---|---|---|---|-------|------------------------|-------|-------|--------|-------|--------|
| Vertebra | L-superior pr. | b | 14 | 15 | 12 | | | 1 | ha; ab; hb1 | 0.15 | 0.1 | 48 | 0.6 | 43 |
| Vertebra | L-transverse pr. | b | 12 | 23 | 8 | | | 0.4 | ha; ab; hb1 | 0.15 | 0.1 | 48 | 0.6 | 43 |
| Vertebra | T- body | c | 27 | 33 | 28 | | | 1.3 | lateral | 0.21 | 0.15 | 48 | 0.5 | 43 |
| Vertebra | T- lamina+inf. | b | 32 | 10.2 | 4.2 | | | 1.3 | ha; ab1 | 0.16 | 0.15 | 48 | 0.6 | 43 |
| Vertebra | T-spinous pr. | b | 10.3 | 50 | 5.1 | | | 1.3 | ha; ab; hb1 | 0.16 | 0.15 | 48 | 0.6 | 43 |
| Vertebra | T-superior pr. | b | 11.4 | 11.3 | 4.4 | | | 1.3 | ha; ab; hb1 | 0.16 | 0.15 | 48 | 0.6 | 43 |
| Vertebra | T-transverse pr. | b | 12 | 18 | 10.6 | | | 1.3 | ha; ab; hb1 | 0.16 | 0.15 | 48 | 0.6 | 43 |

in separation of segments with relatively uniform *Ct.Th* and in the use of the results of morphometry, that are not affected by a method resolution. However, for some sites differences in *Ct.Th* cannot be explained by skeletal segmentation. For example, in UF phantoms *Ct.Th* of newborn vertebrae is comparable to *Ct.Th* of a phantom of a 15-year-old child. In contrast, SPSD phantoms of newborn vertebrae are not covered with a cortical layer at all.

A previous study [19] shows that the AM dose due to incorporated Sr isotopes mainly depends on the segment phantom dimensions, *Ct.Th* and *BV/TV*. Doses to the entire skeleton are the result of weighted averaging of the AM doses of bone segments, considering the site-specific AM fraction (Table 2). By weighted averaging of the parameters that influence dose formation (similar to the dose averaging), we obtain the effective volume of spongiosa, the effective *BV/TV*, and the effective *Ct.Th*. Fig 8 demonstrates the age dependence of the main dose forming parameters.

The age dynamics of the effective parameters depends on two factors: age-related changes in the segment-specific reference parameters and age-related redistribution of AM among skeleton sites. As a result, the age dependences of effective spongiosa volume and *Ct.Th* appear nonmonotonic. The decrease in the effective spongiosa volume after the age of 10 is associated with the termination of hematopoiesis in the distal parts of the tibia and fibula with large spongiosa volume and increase of the hematopoietic importance of the pelvis, which consists of relatively small segments. Some decrease of effective *Ct.Th* at the age of 10 is due to the termination of hematopoiesis in the middle parts of diaphysis of tube bones with thick cortical layers as well as a decrease of the hematopoietic importance of the flat bones of the skull with thick cortex. Effective *BV/TV* decreases steadily with age, reflecting the trend of age-related decrease of *BV/TV*. The highest *BV/TV* in newborns results from the bones of the skull, pelvis, feet, hands and cervical vertebrae. The absolute maximum was observed in the cervical vertebrae of the newborn. The lowest *BV/TV* value was observed for the adult proximal humerus.

Segment-specific dose factors (DFs) for converting radionuclide activity concentrations in trabecular bone into dose rates in AM, DF(AM←TBV), do not depend on spongiosa volume if all linear dimensions exceed 0.42 cm [19,22]. Otherwise, DF (AM←TBV) decreases with decreasing linear dimensions. This could lead to an increase of DF(AM←TBV) with age. However, a decrease of *BV/TV*, which is proportional to spongiosa density, leads to a decrease of DF(AM←TBV) for segments of equal or larger dimensions. In other words, we have two oppositely directed influence factors and one can expect a slight and probably nonmonotonic age dependence of DF(AM←TBV).

DF(AM←CBV) also depends on linear dimensions. For relatively small segments, DF(AM←CBV) increases with decreasing bone size, as well as with an increasing thickness of the cortical layer. For this reason, the exclusion of some small segments from the skeleton model could lead to an underestimation of the DF(AM←CBV). However, the proportion of AM in such parts of the skeleton is very small, so their exclusion cannot significantly affect the DF estimation.

## Conclusions

Digital phantoms are useful for dosimetric modeling of internal and external radiation exposure. A new generation of SPSD models was created for the internal dosimetry of bone-seeking beta emitters, improving upon existing phantoms.

**Table 14. Parameters of bone segment phantoms for a reference adult female; macro-architecture (*h, a, b, c, d, Ct.Th*) and microarchitecture (*Tb.Th, Tb.Sp*) values are given in mm; *BV/TV* is in relative units; variability ($\sigma_{Tb.Th}$ and $\sigma_{Tb.Sp}$) is given in percentages; the shape of a segment is designated as follows: c-cylinder, dc-deformed cylinder, b-box, p-prism with a triangle base, t-tube.**

| Site | Segment | Shape | h | a | b | c | d | Ct.Th | Face, covered by Ct.Th | BV/TV | Tb.Th | $\sigma Tb.Th$ | Tb.Sp | $\sigma Tb.Sp$ |
|---|---|---|---|---|---|---|---|---|---|---|---|---|---|---|
| Clavicle | Acromial shaft | c | 51.5 | 24 | 21 | 10 | 10 | 1.8 | lateral | 0.13 | 0.19 | 22 | 0.8 | 23 |
| Clavicle | Ends | dc | 20 | 24 | 21 | | | 0.6 | lateral | 0.29 | 0.14 | 22 | 0.8 | 23 |
| Clavicle | Sternal shaft | dc | 51.5 | 21 | 12 | 10 | 10 | 1.8 | lateral | 0.13 | 0.19 | 22 | 0.8 | 23 |
| Femur | Neck | c | 30.9 | 29.4 | 23.9 | | | 1.9 | lateral | 0.17 | 0.19 | 22 | 0.78 | 23 |
| Femur | Trochanter area | dc | 34.5 | 58 | 39 | 27 | 27 | 2.3 | lateral | 0.11 | 0.136 | 22 | 0.99 | 23 |
| Humerus | Proximal end | dc | 24.9 | 51.3 | 51.3 | 23.9 | 23.9 | 1.1 | lateral | 0.06 | 0.1 | 22 | 2.37 | 23 |
| Pelvis | Acetabulum | t | 29 | 26 | 21 | | | 0.5 3.6 | h ab1 | 0.19 | 0.13 | 10 | 0.6 | 10 |
| Pelvis | Iliac ala | b | 9.5 | 30 | 30 | | | 1 | ab | 0.19 | 0.13 | 10 | 0.6 | 10 |
| Pelvis | Iliac crest | b | 11 | 30 | 13 | | | 1 | ha; ab1 | 0.19 | 0.13 | 10 | 0.6 | 10 |
| Pelvis | Iliac dorsal segment | b | 19 | 30 | 30 | | | 1 | ab | 0.19 | 0.13 | 10 | 0.6 | 10 |
| Pelvis | Ischium ramus | c | 30 | 34 | 25 | | | 0.5 | h | 0.25 | 0.3 | 10 | 1 | 10 |
| Pelvis | Pubis ramus inferior | dc | 47 | 16 | 22 | 26 | 14 | 0.5 | h | 0.25 | 0.3 | 10 | 1 | 10 |
| Pelvis | Pubis ramus superior (lower) | b | 19 | 11 | 33 | | | 0.7 1.5 | hb; ha1 ab1 | 0.17 | 0.29 | 10 | 1 | 10 |
| Pelvis | Pubis ramus superior (upper) | b | 55.8 | 11 | 13 | | | 0.7 1.5 | ha; hb1 ab1 | 0.17 | 0.29 | 10 | 1 | 10 |
| Ribs | 1, 2 | b | 14 | 30 | 5.5 | | | 0.7 | ha; ab | 0.12 | 0.147 | 22 | 0.82 | 23 |
| Ribs | 11, 12 | b | 9.5 | 30 | 4 | | | 0.7 | ha; ab | 0.12 | 0.147 | 22 | 0.82 | 23 |
| Ribs | 3, 4, 9, 10 | b | 11.3 | 30 | 6 | | | 0.7 | ha; ab | 0.12 | 0.147 | 22 | 0.82 | 23 |
| Ribs | 5.6.7.8 | b | 12.5 | 30 | 6.8 | | | 0.7 | ha; ab | 0.12 | 0.147 | 22 | 0.82 | 23 |
| Sacrum | Ala 3–4 | p | 19 | 18 | 38.5 | 38.5 | | 1.5 | bh; ab1 | 0.15 | 0.1 | 48 | 0.6 | 43 |
| Sacrum | Body 1 | b | 30 | 37.8 | 22.2 | | | 1.5 | ha | 0.15 | 0.1 | 48 | 0.6 | 43 |
| Sacrum | Body 2–3 | b | 45.2 | 28 | 13.8 | | | 1.5 | ha | 0.15 | 0.1 | 48 | 0.6 | 43 |
| Sacrum | Body 4–5 | b | 35 | 28 | 8.5 | | | 1.5 | ha | 0.15 | 0.1 | 48 | 0.6 | 43 |
| Sacrum | Pedicle 1 | c | 13.9 | 23.7 | 15.3 | | | 1.5 | h | 0.15 | 0.1 | 48 | 0.6 | 43 |
| Sacrum | Pedicle 2 | c | 14.2 | 25 | 13.6 | | | 1.5 | h | 0.15 | 0.1 | 48 | 0.6 | 43 |
| Sacrum | Pedicle 3 | c | 13.9 | 18.3 | 13.2 | | | 1.5 | h | 0.15 | 0.1 | 48 | 0.6 | 43 |
| Sacrum | Pedicle 4 | c | 13.9 | 14.5 | 11.2 | | | 1.5 | h | 0.15 | 0.1 | 48 | 0.6 | 43 |
| Sacrum | Sacral ala 1 | b | 30 | 21 | 38.6 | | | 1.5 | ha; ab1 | 0.15 | 0.1 | 48 | 0.6 | 43 |
| Sacrum | Sacral ala 2 | b | 26 | 23 | 22.7 | | | 1.5 | ha | 0.15 | 0.1 | 48 | 0.6 | 43 |
| Scapula | Acromion | b | 8.8 | 48 | 26 | | | 0.8 | ha; hb1; ab | 0.22 | 0.24 | 22 | 0.96 | 23 |
| Scapula | Glenoid | c | 20 | 36 | 26 | | | 0.9 | h | 0.22 | 0.24 | 22 | 0.96 | 23 |
| Scapula | Lateral margin | b | 30 | 5 | 10 | | | 0.8 | ha1; hb | 0.22 | 0.24 | 22 | 0.96 | 23 |
| Skull | Flat bones | b | 5.2 | 30 | 30 | | | 1.3 1.5 | ab1 ab2 | 0.52 | 0.29 | 8 | 0.57 | 15 |
| Sternum | Body | b | 0.9 | 30 | 30 | | | 1.1 | ab | 0.15 | 0.15 | 22 | 1.4 | 23 |
| Sternum | Manubrium | b | 1.1 | 30 | 30 | | | 1.45 | ab | 0.15 | 0.15 | 22 | 1.4 | 23 |
| Vertebra | C- body 3–7 | c | 12 | 16 | 15 | | | 0.3 | h | 0.21 | 0.15 | 48 | 0.5 | 43 |
| Vertebra | Cervical body 2 | b | 19.2 | 14.3 | 17.5 | | | 0.3 | hb | 0.21 | 0.15 | 48 | 0.5 | 43 |
| Vertebra | Cervical lateral 1 | b | 15 | 11.4 | 10.5 | | | 0.3 | ha; ab1 | 0.21 | 0.15 | 48 | 0.5 | 43 |
| Vertebra | L- body | c | 27 | 43 | 32 | | | 1.3 | h | 0.16 | 0.15 | 48 | 0.6 | 43 |
| Vertebra | L- lamina+inf.pr | b | 20.4 | 12.7 | 4.1 | | | 1 | ha; ab1 | 0.15 | 0.1 | 48 | 0.6 | 43 |
| Vertebra | L-spinous pr. | b | 20 | 31 | 6 | | | 0.4 | ha; ab; hb1 | 0.15 | 0.1 | 48 | 0.6 | 43 |
| Vertebra | L-superior pr. | b | 14 | 15 | 12 | | | 1 | ha; ab; hb1 | 0.15 | 0.1 | 48 | 0.6 | 43 |

*(Continued)*

**Table 14.** (Continued)

| Site | Segment | Shape | h | a | b | c | d | Ct.Th | Face, covered by Ct.Th | BV/TV | Tb.Th | σTb.Th | Tb.Sp | σTb.Sp |
|------|---------|-------|---|---|---|---|---|-------|------------------------|-------|-------|--------|-------|--------|
| Vertebra | L-transverse pr. | b | 12 | 23 | 8 | | | 0.4 | ha; ab; hb1 | 0.15 | 0.1 | 48 | 0.6 | 43 |
| Vertebra | T- body | c | 22 | 29 | 26 | | | 1.3 | h | 0.21 | 0.15 | 48 | 0.5 | 43 |
| Vertebra | T- lamina+inf. | b | 32 | 10.2 | 4.2 | | | 1.3 | ha; ab1 | 0.16 | 0.15 | 48 | 0.6 | 43 |
| Vertebra | T-spinous pr. | b | 10.3 | 50 | 5.1 | | | 1.3 | ha; ab; hb1 | 0.16 | 0.15 | 48 | 0.6 | 43 |
| Vertebra | T-superior pr. | b | 11.4 | 11.3 | 4.4 | | | 1.3 | ha; ab; hb1 | 0.16 | 0.15 | 48 | 0.6 | 43 |
| Vertebra | T-transverse pr. | b | 12 | 18 | 10.6 | | | 1.3 | ha; ab; hb1 | 0.16 | 0.15 | 48 | 0.6 | 43 |

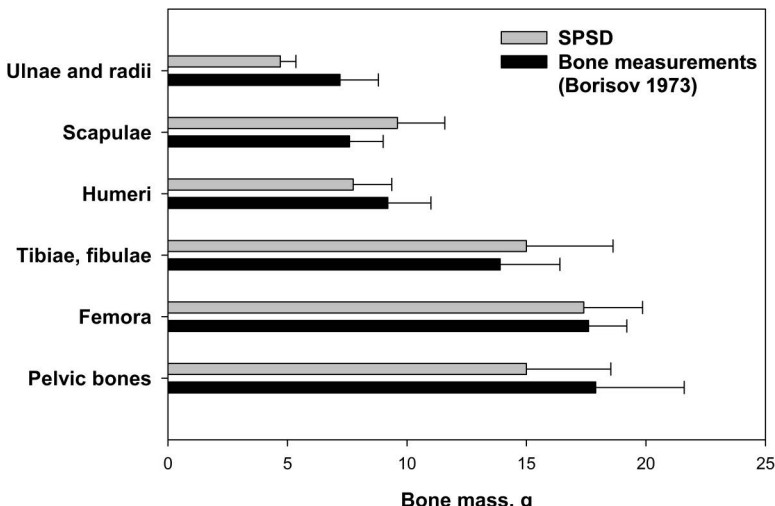

**Fig 5. Comparison of the average bone masses of 40 full-term newborns and SPSD masses of corresponding bone phantoms (calculated as the sum of the masses of the bone segments).** The error bars correspond to ± one standard deviation.

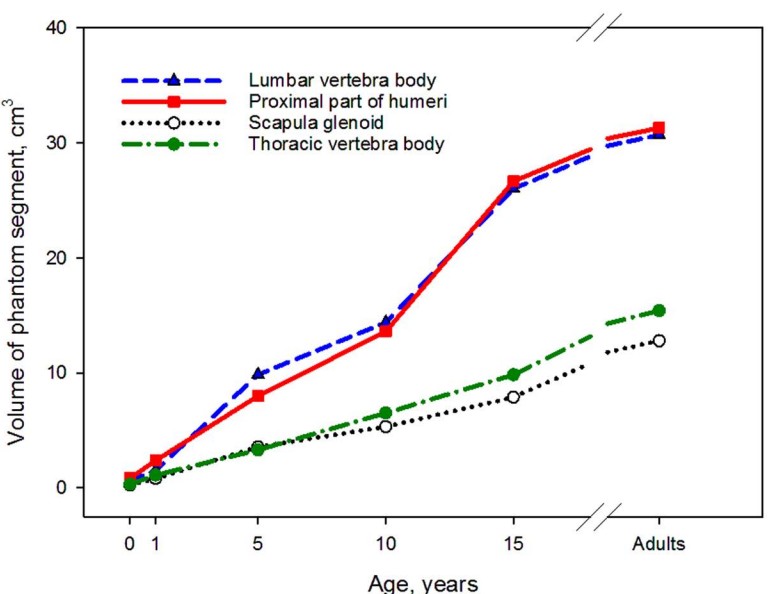

**Fig 6. Age dependence of the total volumes of phantoms of thoracic and lumbar vertebrae, the proximal humerus and scapula glenoid.**

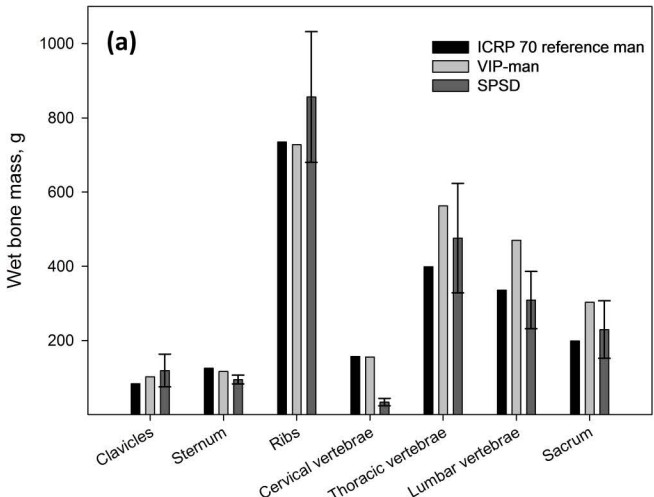
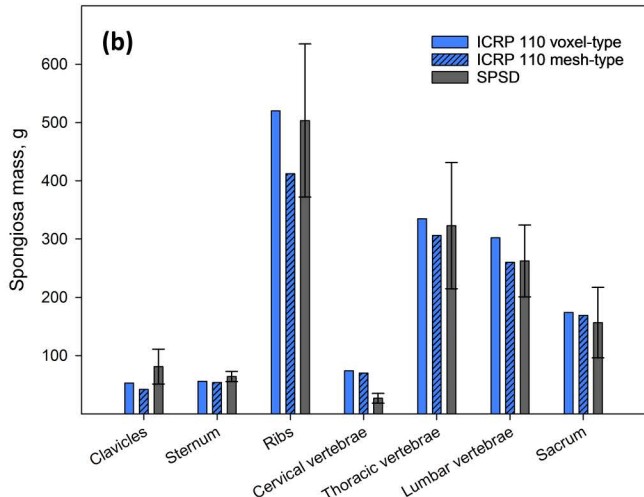

**Fig 7. Comparison of a) total bone masses of the SPSD phantom of adult man, the ICRP 70 reference man and the image-based phantom of VIP-man; and b) spongiosa masses of the SPSD and ICRP 110 voxel and mesh-type phantoms.** The data of SPSD phantoms are shown with error bars corresponding to individual variability (± one standard deviation).

**Table 15. Comparison of *BV/TV* in bone sites described by SPSD and ICRP 70. The rows highlighted in grey correspond to the bone sites for which the value provided by ICRP 70 does not fall within 90% CI of SPSD model.**

| Site | *BV/TV* | |
|---|---|---|
| | ICRP 70 | SPSD Mean [90% CI] |
| Head | 0.55 | 0.52 [0.37–0.7] |
| Scapulae | 0.10 | 0.22 [0.12–0.37] |
| Clavicles | 0.10 | 0.19 [0.11–0.3] |
| Sternum | 0.10 | 0.14 [0.08–0.21] |
| Ribs | 0.10 | 0.12 [0.07–0.19] |
| Cervical vertebrae | 0.12 | 0.21 [0.14–0.13] |
| Thoracic vertebrae | 0.12 | 0.16 [0.09–0.25] |
| Lumbar vertebrae | 0.12 | 0.15 [0.11–0.20] |
| Sacrum | 0.12 | 0.15[0.11–0.20] |
| Os coxae | 0.12 | 0.19 [0.16–0.22] |
| Femora proximal | 0.15 | 0.13 [0.07–0.22] |
| Humeri proximal | 0.15 | 0.06 [0.02–0.12] |

SPSD phantom parameters are based on available published data (including new data) on skeletal morphology of Asians and Caucasians, assumed to be representative of the combined population of interest. The new set of morphometric data (mainly based on bone microarchitecture) has been obtained with advanced measurement methods. Bone-specific masses generally do not contradict ICRP reference values. Bone parameters provided in the paper can be used as an alternative to the ICRP model of reference man.

The models will be integrated into the standardized software for internal dose calculations and be used to produce new AM dose estimates for the residents of the contaminated territories of the Urals. The phantoms can also be used for other dosimetric studies, such as dosimetry of radiopharmaceuticals.

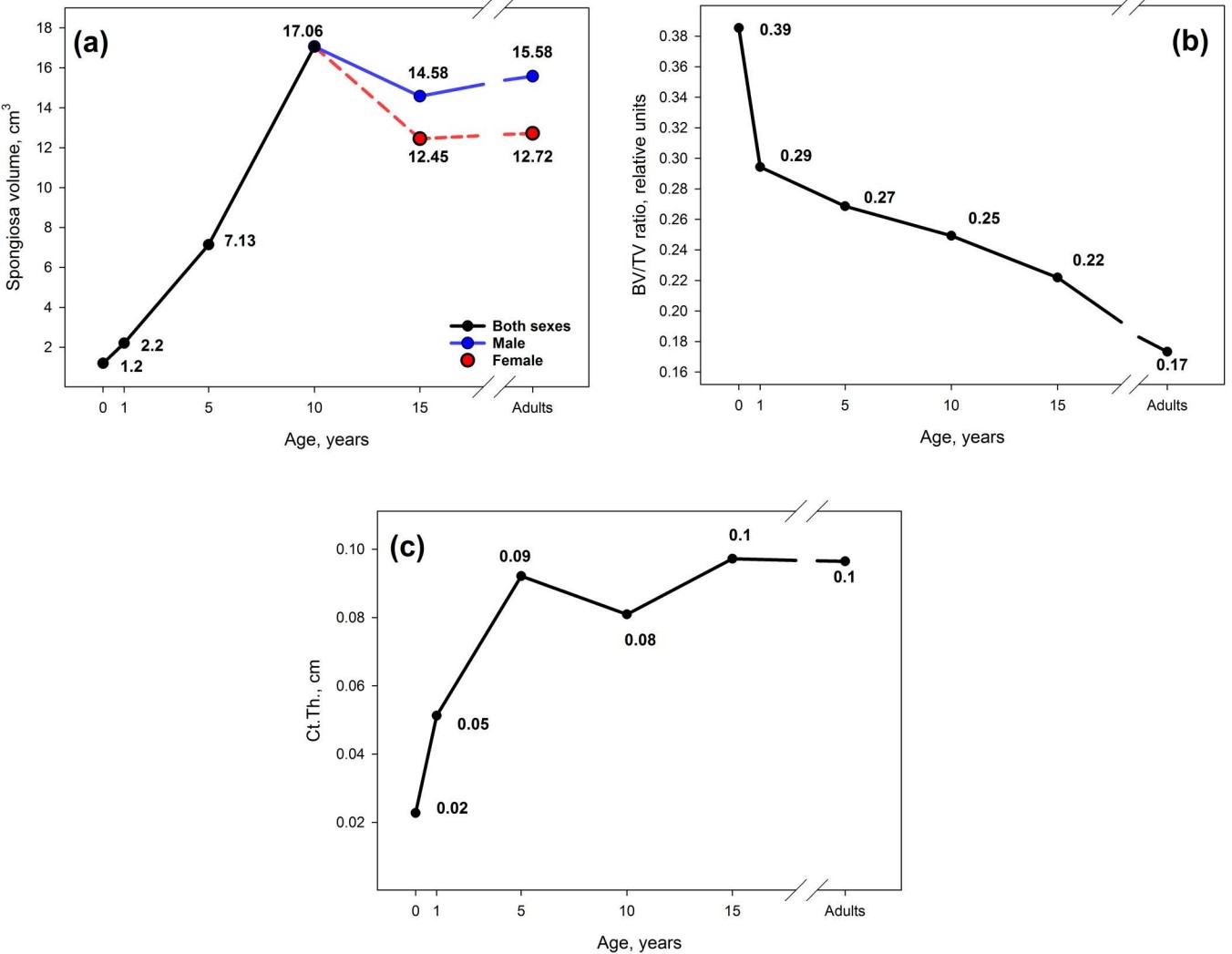

**Fig 8. Effective volume of spongiosa (a), the effective *BV/TV* (b) and the effective *Ct.Th* (c).**

## Supporting information

**S1 File. Tables with coefficients of variation of parameters of SPSD phantoms.**
(DOCX)

**S2 File. Tables with the volumes of simulated media (trabecular bone, cortical bone, and bone marrow) that compose the SPSD phantoms.**
(DOCX)

## Author contributions

**Conceptualization:** Elena A. Shishkina, Evgenia I. Tolstykh.

**Formal analysis:** Pavel A. Sharagin, Evgenia I. Tolstykh.

**Funding acquisition:** Michael A Smith, Bruce A. Napier.

**Investigation:** Pavel A. Sharagin, Michael A Smith.

**Methodology:** Pavel A. Sharagin, Elena A. Shishkina, Evgenia I. Tolstykh.

**Project administration:** Michael A Smith, Bruce A. Napier.

**Software:** Pavel A. Sharagin.

**Supervision:** Michael A Smith, Bruce A. Napier.

**Writing – original draft:** Pavel A. Sharagin, Elena A. Shishkina, Evgenia I. Tolstykh.

**Writing – review & editing:** Pavel A. Sharagin, Elena A. Shishkina, Evgenia I. Tolstykh, Michael A Smith, Bruce A. Napier.

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
