## [Decision Letter · Decision Letter 0]

Dear Dr. Smith,

Thank you for submitting your manuscript to PLOS ONE. After careful consideration, we feel that it has merit but does not fully meet PLOS ONE’s publication criteria as it currently stands. Therefore, we invite you to submit a revised version of the manuscript that addresses the points raised during the review process.

We look forward to receiving your revised manuscript.

Kind regards,

Sakae Kinase, Ph.D.

Academic Editor

PLOS ONE

Journal Requirements:

Additional Editor Comments:

Your paper has been carefully considered by two referees. The comments indicate that some revisions are necessary before the paper can be considered for publication in PLOS One.

These are not Major revisions, and it was determined that a response to the specific comments from Reviewer 2 is necessary. Additionally, regarding the matters that Reviewer 2 mentioned as general comments, further explanation would be not needed, as you have presented this works as a logical extension of the earlier work presented in references 27 and 28.

You have developed SPSD models for internal dose evaluations of the South Urals population. As you also pointed out, many conditions and assumptions are required to develop reference models used for skeletal dosimetry. In your study, a comprehensive set of bone/age-specific models based on data on the skeletal morphology of Asians and Caucasians, are presented that utilize a vast array of data sources on bone macro/micro anatomy. SPSD models that do not use autopsy materials as sophisticated stylized ones, would have a significant advantage compared to other models.

Reviewers' comments:

Reviewer's Responses to Questions

**Comments to the Author**

1. Is the manuscript technically sound, and do the data support the conclusions?

Reviewer #1: Yes

Reviewer #2: Yes

2. Has the statistical analysis been performed appropriately and rigorously?

Reviewer #1: Yes

Reviewer #2: N/A

3. Have the authors made all data underlying the findings in their manuscript fully available?

Reviewer #1: Yes

Reviewer #2: Yes

4. Is the manuscript presented in an intelligible fashion and written in standard English?

Reviewer #1: Yes

Reviewer #2: Yes

Reviewer #1: The submitted paper presents an array of sophisticated stylized models of individual bones of the human skeletal across a range of ages - newborn to adult - for the purpose of assessing active marrow dosimetry for skeletal-seeking radionuclides. The stated purpose of the models is to support ongoing radiation epidemiological studies of radiation-induced leukemia risk in the cohorts of the Ural Mountains to include the members of the Techa River villagers of the former USSR. The work presented is a logical extension of the earlier work presented in Ref. 27 and 28. In this paper, a comprehensive set of bone/age-specific models are presented that utilize a vast array of data sources on bone macro/micro anatomy. The paper is well written and presented, and is a welcomed additional to the literature in bone dosimetry models and applications.

Reviewer #2: I am happy to have had the opportunity to review this. Please see the attached file.

General comment

This is a study on improving the dose evaluation model to avoid underestimating or overestimating the dose of beta rays to the bone marrow in internal exposure in typical individuals. It is thought that the study is also trying to answer questions from the public(KOBAYASHI 2019)(Kobayashi 2024).

Since underestimating the dose to the bone marrow leads to underestimating the risk for the exposed individual, and overestimating the dose leads to underestimating the risk in epidemiological studies, it would be better if the study also mentioned the impact it would have on radiation protection.

From the perspective of promoting the significance of the research, why not also explain the development from Reference 28?

**Do you want your identity to be public for this peer review?** For information about this choice, including consent withdrawal, please see our Privacy Policy

Reviewer #1: No

Reviewer #2: No

---

## [Author Response · Author response to Decision Letter 1]

26 May 2025

Our response to reviewer comments has been uploaded and is labeled as "Response to Reviewers" with filename: Response to Reviewers 20250518.docx.

---

## [Decision Letter · Decision Letter 1]

Dear Dr. Smith,

Thank you for submitting your manuscript to PLOS ONE. After careful consideration, we feel that it has merit but does not fully meet PLOS ONE’s publication criteria as it currently stands. Therefore, we invite you to submit a revised version of the manuscript that addresses the points raised during the review process.

We look forward to receiving your revised manuscript.

Kind regards,

Sakae Kinase, Ph.D.

Academic Editor

PLOS ONE

Journal Requirements:

Additional Editor Comments :

Your revised paper has been carefully considered by two referees. The referee's comment is attached. Please carefully consider the comment and resubmit, as soon as possible, an amended version of the paper, including figures.

Reviewers' comments:

Reviewer's Responses to Questions

**Comments to the Author**

Reviewer #1: All comments have been addressed

Reviewer #2: (No Response)

2. Is the manuscript technically sound, and do the data support the conclusions?

Reviewer #1: Yes

Reviewer #2: Yes

3. Has the statistical analysis been performed appropriately and rigorously?

Reviewer #1: Yes

Reviewer #2: N/A

4. Have the authors made all data underlying the findings in their manuscript fully available?

Reviewer #1: Yes

Reviewer #2: Yes

5. Is the manuscript presented in an intelligible fashion and written in standard English?

Reviewer #1: Yes

Reviewer #2: Yes

Reviewer #1: The authors were responsive to the prior review of their paper.

The authors were responsive to the prior review of their paper.

The authors were responsive to the prior review of their paper.

The authors were responsive to the prior review of their paper.

Reviewer #2: Thank you very much for your kind response. I was impressed by the sincere attitude of the authors.

Please see the attached file.

**Do you want your identity to be public for this peer review?** For information about this choice, including consent withdrawal, please see our Privacy Policy

Reviewer #1: No

Reviewer #2: **Yes: ** Ichiro YAMAGUCHI

---

## [Author Response · Author response to Decision Letter 2]

15 Jun 2025

June 15, 2025: Response to reviewer comments is provided in a file included today with upload: "Response to Reviewers 20250615".

---

## [Editor Report · Decision Letter 2]

Stochastic parametric skeletal dosimetry model for humans: Pediatric and adult computational skeleton phantoms for internal bone marrow dosimetry

PONE-D-25-02823R2

Dear Dr. Smith,

We’re pleased to inform you that your manuscript has been judged scientifically suitable for publication and will be formally accepted for publication once it meets all outstanding technical requirements.

Kind regards,

Sakae Kinase, Ph.D.

Academic Editor

PLOS ONE

Additional Editor Comments (optional):

I have much pleasure in recommending this paper for publication in PLOS One. The manuscript has been substantially with changes according to reviewers' comments.
---

## [Editor Report · Acceptance letter]

PONE-D-25-02823R2

PLOS ONE

Dear Dr. Smith,

I'm pleased to inform you that your manuscript has been deemed suitable for publication in PLOS ONE. Congratulations! Your manuscript is now being handed over to our production team.

Kind regards,

on behalf of

Professor Sakae Kinase

Academic Editor

PLOS ONE